# Less is more? Ultra-low carbohydrate diet and working dogs' performance

**Arnon Gal** [1]⊕*, **Williams Cuttance**[2], **Nick Cave**[3], **Nicolas Lopez-Villalobos**[4], **Aaron Herndon** [5], **Juila Giles**[6], **Richard Burchell**[7]⊕

**1** Department of Veterinary Clinical Medicine, College of Veterinary Medicine, University of Illinois at Urbana-Champaign, Urbana, Illinois, United States of America, **2** VetEnt Te Kuiti, Hamilton, New Zealand, **3** School of Veterinary Science, Massey University, Palmerston North, New Zealand, **4** School of Agriculture and Environment, Massey University, Palmerston North, New Zealand, **5** School of Veterinary Science, The University of Queensland, Gatton, Queensland, Australia, **6** Totally Vets, Feilding, New Zealand, **7** North Coast Veterinary Specialist and Referral Centre, Sunshine Coast, Queensland, Australia

⊕ These authors contributed equally to this work.
\* agal2@illinois.edu

**Data Availability Statement:** All relevant data are used for the statistical analyses and for the tables and figures are available within the following stable data repository (doi: 10.17632/d76znr6jvk.1).

## Abstract

New Zealand farm working dogs are supreme athletes that are crucial to agriculture in the region. The effects that low or high dietary carbohydrate (CHO) content might have on their interstitial glucose (IG) and activity during work are unknown. The goals of the study were to determine if the concentration of IG and delta-g (a measurement of activity) will be lower in dogs fed an ultra-low CHO high fat diet in comparison to dogs fed a high CHO low fat diet, and to determine if low concentrations of IG are followed by reduced physical activity. We hypothesized that feeding working farm dogs an ultra-low CHO diet would reduce their IG concentrations which in turn would reduce physical activity during work. We prospectively recruited 22 farm dogs from four farms. At each farm, dogs were randomized to one of two diets and had a month of dietary acclimation to their allocated diet. The macronutrient proportions as a percentage of metabolizable energy (%ME) for the high CHO low fat diet (Diet 1) were 23% protein, 25% fat, and 52% CHO, and for the ultra-low CHO high fat diet (Diet 2) 37% protein, 63% fat, and 1% CHO. Following the acclimation period, we continuously monitored IG concentrations with flash glucose monitoring devices, and delta-g using triaxial accelerometers for 96 h. Dogs fed Diet 2 had a lower area under the curve (±SE) for IG (AUC $_{Diet\ 2}$ = 497 ± 4 mmol/L/96h, AUC $_{Diet\ 1}$ = 590 ± 3 mmol/L/96h; $P$ = 0.002) but a higher area under the curve (±SE) for delta-g (AUC $_{Diet\ 2}$ = 104,122 ± 6,045 delta-g/96h, AUC $_{Diet\ 1}$ = 80,904 ± 4,950 delta-g/96h; $P$< 0.001). Interstitial glucose concentrations increased as the activity level increased ($P$ < 0.001) and were lower for Diet 2 within each activity level ($P$ < 0.001). The overall incidence of low IG readings (< 3.5 mmol/L) was 119/3810 (3.12%), of which 110 (92.4%) readings occurred in the Diet 2 group ($P$ = 0.001). In the Diet 2 group, 99/110 (90%) of the low IG events occurred during the resting period (19:00–06:00). We conclude that feeding Diet 2 (ultra-low CHO high fat diet) to working farm dogs was associated with increased delta-g despite decreased IG concentrations. Interstitial glucose concentrations were positively associated with dogs' activity levels independent of diet. Lastly, events of low IG occurred at a low incidence and were predominantly seen between 19:00–06:00 in dogs fed the ultra-low CHO high fat diet.

**Funding:** AG received funding for this study from the Working Dog Centre, Massey University, New Zealand (http://workingdogs.massey.ac.nz/). The funders had no role in study design, data collection and analysis, decision to publish, or preparation of the manuscript.

**Competing interests:** The authors have declared that no competing interests exist.

## Introduction

There are approximately 200,000 working farm dogs across New Zealand [1]. These dogs are supreme athletes that work in all weather conditions and might run 60–100 km/d on steep terrain, often at 20–30 km/h [1]. Hence, adequate nutrition in the face of high-energy expenditure is paramount to the general health and performance of these dogs. There are currently two common feeding practices for working farm dogs in New Zealand that differ in their dietary macronutrient content: Ultra-low CHO, high fat sources and high CHO, low fat sources of metabolic energy (%ME). However, the effect of dietary macronutrient content on the activity of New Zealand's working farm dogs is hitherto unknown. Dogs do not require dietary CHO, and exercising sled dogs may have improved performance when fed a high-fat CHO-free diet [2]. However, the concept of "cross-over", where muscle utilization of CHO for ATP production increases with increased exercise intensity, has been demonstrated in several different mammalian species, including dogs [3]. Since the intensity of activity engaged by working farm dogs in New Zealand is likely to be greater than the sustained moderate-intensity endurance activity of sled dogs, New Zealand's working farm dogs may be unable to maintain normal body glucose levels when fed a low CHO diet.

One aspect of nutrition is the maintenance of adequate glycemic control. Blood glucose during the fasted and absorptive phases is derived from internal and external sources, respectively. In the fasted state, glycogenolysis of hepatic glycogen stores and gluconeogenesis maintain euglycemia [4, 5]. In contrast, in the absorptive phase, dietary glucose is directly utilized to maintain euglycemia, and excess glucose is converted to storage fuels (i.e., fat and glycogen) as long as the diet has sufficient CHO content [6]. Prolonged physical activity can deplete hepatic and muscle glycogen stores, and maintenance of glucose levels would rely on hepatic gluconeogenesis. However, under conditions of strenuous and prolonged physical exercise, gluconeogenesis might fail to maintain euglycemia. Therefore, feeding diets with high CHO content to dogs that are undergoing strenuous and prolonged physical exercise might support the maintenance of glucose levels for a longer time through maximizing muscle and hepatic glycogen stores and rapidly metabolizable CHO absorption from the gastrointestinal tract.

Healthy human endurance athletes occasionally experience hypoglycemic episodes during exercise [7]. Therefore, hypoglycemia might also occur in working farm dogs and could negatively impact their performance, and general well-being, thus justifying an investigation. This study tested the hypothesis that working farm dogs fed an ultra-low CHO diet would reduce their body glucose levels which in turn would reduce physical activity during work. (Fig 1). The study aims were to determine if IG and delta-g (a measurement of activity) would differ between dietary groups, and if reduced delta-g would follow periods of low levels of IG.

## Materials and methods

### Study design

This prospective, randomized, controlled experimental field study was approved by the Massey University Animal Ethics Committee (Protocol 16/02). We recruited 22 dogs from four farms on the North Island of New Zealand and randomized the dogs into two balanced groups with respect to diet (*n* = 11 each) using an online randomization tool (www.randomizer.org); there was no attempt to balance the dogs per age, sex, breed, or bodyweight during the randomization and recruitment (Tables 1 and 2). Dogs were recruited after receipt of each owner's verbal consent to participate in the study. Dogs were only included in the study if they had been assessed by a physical exam by at least one of the authors and by the history provided by the owner and were deemed to be in good health. Acclimation of the dogs to the two diets started

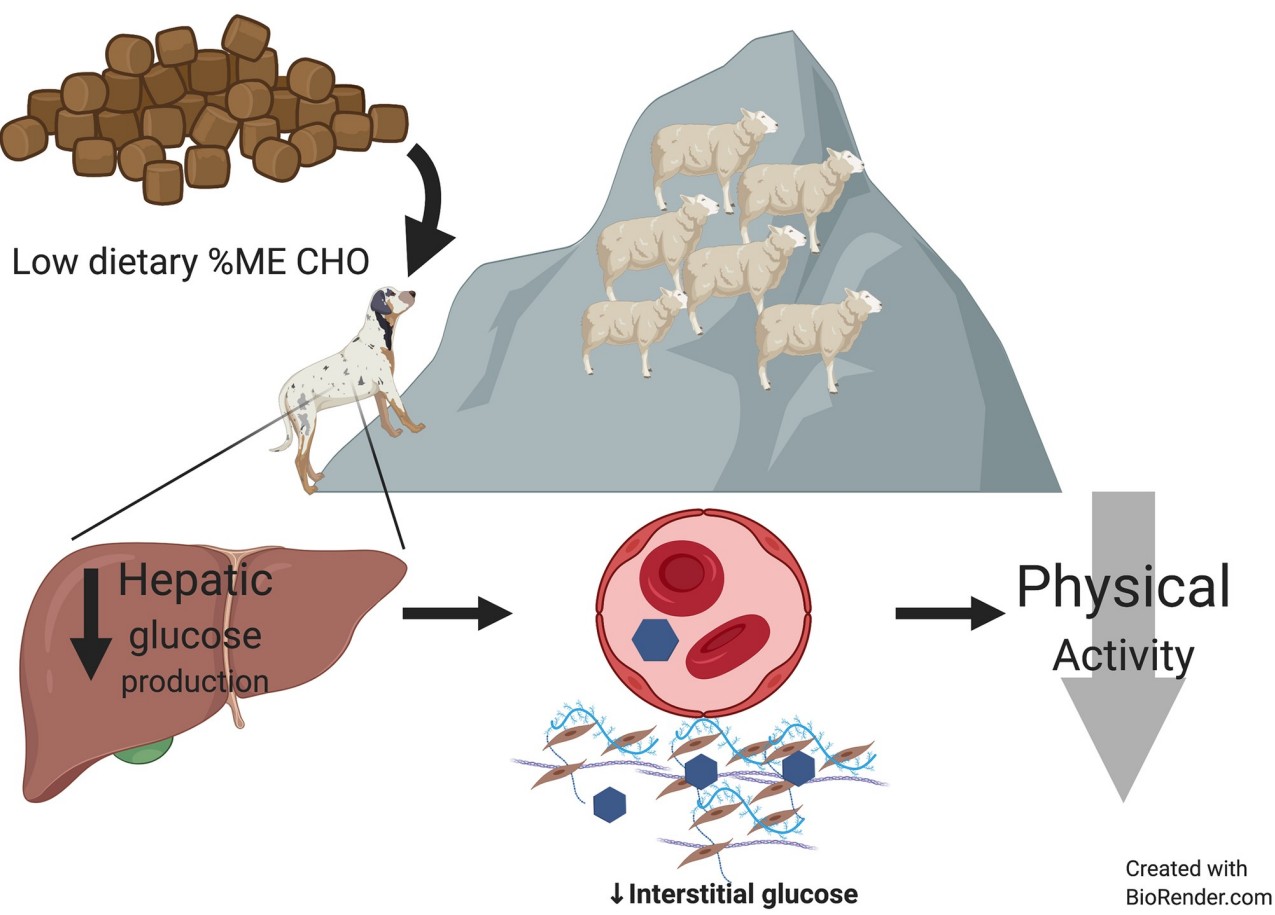

**Fig 1. Hypothesis model.** We hypothesized that an ultra-low CHO high fat diet would reduce IG through decreased hepatic glucose production resulting in decreased delta-g (a measurement of activity). %ME CHO, % of metabolic energy from carbohydrate sources.

**Table 1. Demographics and descriptive statistics of the dog cohort in this study.**

| Farms | All | A | B | C | D |
|---|---|---|---|---|---|
| Number of dogs | 22 | 5 | 5 | 8 | 4 |
| Age (years)[#] | 3.5 (4) | 6[a,b] (3.5) | 2[a] (1) | 4.5[b] (3.5) | 3[a,b] (1) |
| BW (kg) at study entry[†] | 25.2±2.7 | 20.7±3.2 | 25.4±7.8 | 28.4±7.8 | 24±4.4 |
| BW (kg) at the beginning of 96 h[†] | 24.7±6.3 | 21.8±2.8 | 24.3±7.7 | 26.9±7.8 | 24.2±4.1 |
| BW (kg) at the end of 96 h[†] | 24.7±6.7 | 20.9±2.7 | 25±8.3 | 27.5±7.9 | 23.4±4.5 |
| Males, Females | 16, 6[*] | 2, 3 | 4, 1 | 8, 0 | 2, 2 |
| Heading dog, Huntaway, Mixed breed, Border Collie[¶] | 9, 9, 3, 1 | 2, 2, 1, 0 | 3, 1, 0, 1 | 4, 4, 0, 0 | 0, 2, 2, 0 |

**BW**, bodyweight;

[†], results are presented as mean (±SD);

[#], results are presented as median (IQR); different superscript letters represent a significant difference ($P < 0.050$) between farms;

[¶], significant under-represented breed ($P < 0.001$);

[*], significant difference ($P < 0.050$) between males to females; All dogs, but one intact female, were neutered.

**Table 2. Demographics and descriptive statistics per diets at the beginning of the acclimation period.**

|  | Diet 1 | Diet 2 | *P* value |
|---|---|---|---|
| #Baseline age [median (IQR); year] | 3.5 (3.5) | 3.5 (3.5) | 0.947 |
| *Baseline body weight (mean ± SD; kg) | 23.6 ± 4.5 | 26.5 ± 8.1 | 0.850 |
| #Baseline body condition score [median (IQR); 1–9 scale] | 4 (2) | 4 (1) | 0.409 |
| +Sex (M, F) | 9M, 2F | 7M, 4F | 0.338 |
| +Farms (A, B, C, D) | 2A, 2B, 5C, 2D | 3A, 3B, 3C, 2D | 0.825 |
| +Breeds (Bc, Hd, Hw, Mx) | 1Bc, 5Hd, 3Hw, 2Mx | 0Bc, 4Hd, 6Hw, 1Mx | 0.485 |

**Bc**, Border collie; **CHO**, carbohydrates; **F**, female; **Hd**, Heading dog; **Hw**, Huntaway dog; **IQR**, interquartile range; **M**, male; **Mx**, mixed breed dog; **SD**, standard deviation.

one month prior to the commencement of each of the three 96 h study periods, which took place during times of peak seasonal work activity, during which the dogs in each farm worked together as a team. Because of the difficulties in timing the dietary acclimation to match with the unpredictable timing of the periods of peak seasonal work activity (which depends on appropriate weather conditions), the study included three separate 96 h periods between October and November. We recorded the dogs' bodyweights at recruitment, and at the beginning and at the end of each of the study's 96 h periods, during which, we continuously monitored IG levels using a flash glucose monitor (FreeStyle Libre, Abbott Diabetes Care, Doncaster, VIC, Australia). Dogs were also fitted with a triaxial accelerometer (TAA; Heyrex®, Wellington, New Zealand) which was fixed to their neck collar, to continuously quantify their delta-g.

## Diets

The macronutrient proportions as a percentage of metabolizable energy (%ME) for the first diet (Diet 1) were 23% protein, 25% fat, and 52% CHO (Pedigree Adult real chicken®, MARS Petcare New Zealand, Auckland, New Zealand), and for the second diet (Diet 2), 37% protein, 63% fat, and 1% CHO (Chicken Feast Freeze Dried®, K9 Natural, Christchurch, New Zealand). The ingredients and nutritional analysis of the two diets are presented in S1 and S2 Tables. During the acclimation and the 96 h testing periods, the dogs were fed their allocated diet exclusively once daily between 6pm-8pm whilst in their individual kennels. No other source of food was allowed. The dogs had ad libitum access to water. The owners were asked to feed the dogs during the acclimation period to maintain bodyweight similar to other studies in athletic dogs [8, 9]. The empirically determined amount fed by the end of the acclimation period was then maintained throughout the 96 h test periods. We based our rationale for choosing the "feeding to maintain body-weight" strategy for several reasons. Firstly, given the nature of the study design, we were unable to quantify the dietary intake of individual dogs. Secondly, we expected large variations in energy expenditure that would lead to significant differences in bodyweight if fed to a predetermined intake. Thirdly, large variations in maintenance energy requirements between dogs, even of the same bodyweight, have been previously documented [10]. Hence, feeding to maintain bodyweight seemed the most appropriate approach.

## Sample collection and handling

**Blood collection.** Whole blood samples (5 mL) were collected by jugular venipuncture from each of the dogs enrolled in the study between 8am-10am after an overnight fast, and split between serum (BD Vacutainer, Auckland, New Zealand) and sodium fluoride (BD

Vacutainer, Auckland, New Zealand) blood collection tubes. The samples were collected following dietary acclimation and at the beginning and at end of the 96 h study periods. Whole blood in the serum tubes was allowed to clot for 10 minutes whereas the whole blood in the sodium fluoride tubes was inverted several times to allow for the anticoagulant to mix with the blood. Then, the samples were stored on ice as long as the veterinarian was in the field and shipped on ice overnight to the principal investigator.

**Sample handling and processing.** Immediately upon receipt of the blood samples from the field veterinarian, the blood samples were centrifuged at $3000 \times g$ for 15 min at 4˚C to facilitate harvesting the serum and plasma for insulin and glucose assays, respectively. A local veterinary diagnostic laboratory (IDEXX New Zealand, Palmerston North, New Zealand) measured plasma glucose concentration (OSR6121, Beckman-Coulter, Sydney, NSW, Australia) on an AU680 Chemistry Analyzer (Beckmancoulter, Sydney, NSW, Australia); whereas serum samples for insulin quantification were stored in -80˚C until the end of the study and then shipped on dry ice to the University of Queensland (Gatton, QLD, Australia) for analysis.

## Flash glucose monitoring and triaxial accelerometry

The FreeStyle Libre® (Abbott Diabetes Care, Doncaster, VIC, Australia) was used to measure the IG in the subcutaneous adipose tissue every 15 minutes. To apply the sensor, a 5 cm × 5 cm square on the lower right side of the neck was clipped, and aseptically cleaned. The FreeStyle Libre® sensor was deployed on the prepared area using the pre-loaded applicator and secured with an elastic bandage. The FreeStyle Libre® has been previously shown to have a good correlation with plasma glucose in dogs [11–14]. At the same time, the dogs were also fitted with a triaxial accelerometer (Heyrex®, Wellington, New Zealand) that generates a numerical measurement unit of activity (delta-g) [15–17]. Acceleration in all three axes was summed to produce the delta-g value, which was analyzed as 15 min epochs. The duration of the period of continuous glucose and activity recording was 96 h. During that time, the dogs in each farm worked according to their usual regimen together as a team.

Accelerometers and flash glucose monitors were fitted in all dogs in the morning (between 8am–10am) on the first day of data collection. All data points for IG had a corresponding time-stamped, matched activity datapoint. This permitted the evaluation of the temporal relationship between IG and activity as these could be tracked on a time-matched basis for both IG and activity. To standardize the Time 0 point between dogs, the dogs were normalized based on their delta-g. Hence, the first IG data point that was used for analysis (i.e., Time 0) corresponded to the first nadir of delta-g (i.e., the lowest value for a 15-min epoch) that followed the first recorded peak of delta-g (i.e., the highest value for a 15-min epoch) for each dog. That way, all dogs were aligned based on the elapsed time from Time 0 to the end of the 96 h period of data acquisition.

The measurement of IG in dogs is a relatively new practice, and hence well-established references intervals for canine IG are not available. By way of extrapolation from established blood glucose levels [18], we defined low IG as any IG concentration below 3.5 mmol/L.

## Insulin assay, and the homeostatic model assessment of insulin resistance

Serum samples were maintained at -80˚C until thawing for the assay. Insulin was measured at a regional veterinary reference laboratory (QML Vetnostics Laboratory, Murarrie, QLD, Australia) that participates in the European Society of Veterinary Endocrinology Quality Assurance Program. The reference laboratory reported quality control results for the assay of less than 1 SD from the mean across all participating laboratories in the European Society of Veterinary Endocrinology Quality Assurance Program. A direct chemiluminescent human assay,

which is based on a two-site sandwich immunoassay using a mouse monoclonal capture antibody (ADVIA Centaur Insulin Assay, Siemens-Australia, Bayswater, VIC, Australia) was used. The assay's lower limit of detection and range are 0.5 μU/L and 1–300 mU/L, respectively. The within run coefficient of variation (CV) at insulin concentrations of 27 mU/L and 65 mU/L are 1.05% and 3.9%, respectively. The between run CV at insulin concentrations of 27 mU/L and 65 mU/L are 4.07% and 6.83%, respectively.

The homeostatic model assessment of insulin resistance (HOMA-IR) is an epidemiological tool used in humans to assess insulin resistance. High HOMA-IR levels indicate an increased resistance to insulin, and lower levels indicate an increased sensitivity to insulin's action. The HOMA-IR was calculated as previously described [19, 20] according to the following formula: plasma insulin x serum glucose / 22.5.

## Dog activity levels

We split the dogs' delta-g values over the 96 h period into three levels according to the individual dog's triaxial accelerometer delta-g values as follows: high ($\geq$75% percentile; a total of 981 15-min epochs), moderate (>25% and <75% percentiles; a total of 1803 15-min epochs), and low ($\leq$25% percentile; a total of 915 15-min epochs).

## The 'working period' and 'resting period'

For the purpose of comparing the effect of the diets on IG at the end of the working day (when the dogs rested in the kennel) compared with the period of time of the day where the dogs had been working, we divided the day into two periods: the period spanning between 19:00–06:00 (i.e., 'resting period'; the end of the working day to the beginning of the next working day) and the period from 06:00–19:00 ('working period'). We assumed that during the 'resting period' the dogs spent most of their time sleeping. This assumption was based on 1) the time from sunset to sunrise in Auckland, North Island of New Zealand (https://www.gaisma.com/en/location/auckland.html); and 2) the lower activity quartiles between 19:00–06:00 accounted for 85.6% of the total lower activity quartile over the 96 h periods (S1 Fig).

## Power-sample size and statistical analysis

A priori power analysis (G*Power software version 3.1.9.2) indicated that 9 dogs in each group will be sufficient to detect a difference between two dependent IG means of 100 mg/dL and 70 mg/dL with SD of 30 mg/dL, power of 0.8, alpha error probability of 0.05, and a correlation coefficient of 0.6 between measurements on the same dog. To account for potential losses of dogs during the study period (i.e., dislodgement of the sensor), we recruited 11 dogs per group.

The statistical analysis was performed with the statistical package SAS University Edition (SAS Institute Inc., Cary, NC, USA). The variables in the dataset were examined for normal distribution by inspection of Q-Q plots, histogram, and by the Shapiro-Wilk test. Descriptive statistics were calculated by the MEANS procedure. The median (IQR) were used to describe the variables 'age' and 'body condition score' (Tables 1 and 2). The mean (±SD) were used to describe 'bodyweight at the beginning of the acclimation period', 'bodyweight at the beginning of the 96 h data recording period', and 'bodyweight at the end of the 96 h data recording period' (Table 1). Analysis of variance for the dependent variable IG was performed with the MIXED procedure using a linear mixed model for repeated measures taken on the same dog during the 96 h of data recording [21]. The model included the fixed effects of 'period', 'activity level', 'diet', interaction between 'activity level' and 'diet', and the random effect of 'dog' to account for repeated measures on the same 'dog'. The time (i.e., the 96 h of data recording)

was converted to the class variable 'period' defined with 8 levels. The effects of 'activity level' and 'diet' on the least squares means (LSM) of dependent variable IG were compared by the Fisher's Least Significant Difference test as implemented in the LSMEANS option.

The NPAR1WAY procedure was used to analyze the dependent variables 'age' and 'body condition score' (scale of 1–9) with respect to farm (Table 1) and diet (Table 2) allocation, as these variables did not follow a normal distribution. The NPAR1WAY procedure was also used to analyze the dependent variables 'plasma insulin' and 'HOMA-IR' between the beginning and the end of the 96 h data recording period, as these variables did not follow a normal distribution. Analysis of variance for the dependent variables 'bodyweight at the beginning of the acclimation period', 'bodyweight at the beginning of the 96 h data recording period', and 'bodyweight at the end of the 96 h data recording period' was performed with the MIXED procedure using a linear mixed model. The model included the fixed effects of farm (Table 1) and Diet (Table 2).

Repeated measures of 'bodyweight' on the same dog were analyzed using the MIXED procedure with a mixed model that include the fixed effect of 'diet', 'time' (i.e., the beginning of the acclimation period, and the beginning and the end of the 96 h data recording period), and interaction between 'diet' and 'time', and the random effect of 'dog' to account for repeated measures on the same dog. Repeated measures of 'serum glucose' on the same dog were analyzed using the MIXED procedure with a mixed model that include the fixed effect of 'diet', 'time' (i.e., the beginning and the end of the 96 h data recording period), and interaction between 'diet' and 'time', and the random effect of 'dog' to account for repeated measures on the same dog.

Analysis of the frequency of low IG readings during the 'working period' and 'resting period', between activity levels, and between diets was performed with Chi-Square Test as it is implemented in the FREQ procedure.

Curves for 'IG' and 'delta-g' during the period of recorded data were modelled using a mixed model fitting nine-knots splines for each dog using the statistical software ASReml [22, 23]. The model included the fixed effect of 'diet' and random effect of 'dog'. A spline for 'IG' and 'delta-g' were modelled for each dog and the best predicted curve was estimated for each diet. The differences between the areas under the curves (AUC) for 'IG' and 'delta-g' were performed by calculating their z statistics according to the following formula: $z = |AUC_1 - AUC_2| / \sqrt{SE_1^2 + SE_2^2}$. Fitting a random regression model with ASReml, modeling the 'IG' and 'delta-g' for each dog, and choosing the best predicted AUC for each diet was deemed the most accurate way to estimate the effect of diet on 'IG' and 'delta-g'. The unfitted raw data is presented in S2 Fig.

## Results

### The effect of the diets on bodyweight

At the beginning of the acclimation period there were no significant differences between the dogs in diet groups 1 and 2 with regards to 'age', 'bodyweight', 'body condition score', 'sex', distribution within farm, and breeds (Table 2).

The analysis of variance for bodyweight indicated that Dogs fed Diet 1 maintained their bodyweight from the beginning of the acclimation period to the beginning of the 96 h study period (LSM ± SE 23.6 ± 2.1 kg vs. 23.9 ± 2.1 kg; $P$ = 0.916), whilst dogs fed diet 2 lost approximately 1.2 kg (LSM ± SE 26.5 ± 1.9 kg vs. 25.3 ± 1.9 kg; $P$ = 0.007). The bodyweights of all dogs remained unchanged during the 96 h study period (Diet 1, LSM ± SE 23.9 ± 21. kg vs. 23.9 ± 21. kg, $P$ = 1.000; Diet 2, LSM ± SE 25.3 ± 1.9 kg vs. 25.3 ± 1.9 kg, $P$ = 1.000).

## The effect of diets on activity level and interstitial glucose concentration

The predicted areas under the curve for 'IG' and 'delta-g' per diet are graphically depicted in Fig 2. Dogs fed diet 2 had a significantly lower area under the curve (±SE) for IG when compared with the Diet 1 (AUC $_{Diet\ 2}$ = 497 ± 4 mmol/L/96 h, AUC $_{Diet\ 1}$ = 590 ± 3 mmol/L/96 h; $P$ = 0.002). In contrast, dogs fed diet 2 had a significantly higher area under the curve (±SE) for delta-g when compared with the Diet 1 (AUC $_{Diet\ 2}$ = 104,122 ± 6,045 delta-g/96 h, AUC $_{Diet\ 1}$ = 80,904 ± 4,950 delta-g/96 h; $P$ < 0.001).

## Interstitial glucose concentration as a function of three activity levels

The LSM (±SE) IG was significantly greater with higher activity level ($P$ < 0.001), and in Diet 1 ($P$ < 0.001; Fig 3). In addition, there was a significant interaction between the level of activity and diet ($P$ < 0.001), whereby dogs fed Diet 2 had a greater increase in IG with increased activity than dogs fed Diet 1 (Fig 3).

## The effect of diet, and the 'working period' and 'resting period' on the incidence of low levels of IG

Over the three 96 h periods, there were 3810 IG readings; 2090 (55%) during the 'working period', and 1720 (45%) during the 'resting period'. Overall, the proportion of low IG readings (< 3.5 mmol/L) was 119/3810 (3.12%; S3 Fig). Despite the lower total number of IG readings during the 'resting period' relative to the 'working period', a significantly ($P$ < 0.001) higher proportion of low IG readings occurred during the 'resting period' (101/119; 84.9%) relative to the 'working period' (18/119; 15.1%; S4 Fig).

There were 1504/3810 and 2306/3810 IG readings from dogs in Diet 1 and Diet 2 groups, respectively ($P$ < 0.001). We found that 9/119 (7.6%) of the low IG readings occurred in dogs from Diet 1 group (0.6% of the total IG readings for this group), and 110/119 (92.4%) in dogs from Diet 2 group (4.8% of the total IG readings for this group; $P$ = 0.001).

We observed a reciprocal pattern in the timing of the low IG readings between the two diets. Seven of the 9 (77.8%) low IG readings in Diet 1 group occurred during the 'working period' and only 2/9 (22.2%) occurred during the 'resting period'. In contrast, 99/110 (90%) of the low IG readings in the Diet 2 group occurred during the 'resting period' and only 11/110 (10%) occurred during the 'working period'. There was a significant interaction between the diet and the 'resting period' / 'working period' with regards IG ($P$ < 0.001).

Overall, 10/22 (45.5%) dogs experienced low IG events (median of 6.5; IQR 17.5; range 1–42). Eight of the 10 dogs (80%) were from Diet 2 group and accounted for 110/119 of the low IG readings, whereas 2/10 dogs were from Diet 1 group and accounted for 9/119 of the low IG readings. Two of the 10 dogs had a total of two low IG readings during the 'working period' without any accompanying low IG readings during the 'resting period'. Similarly, 2/10 dogs had seven low IG readings during the 'resting period' without any accompanying low IG readings during the 'working period'. The other 6/10 dogs had low IG readings both during the 'resting period' and during the 'working period'.

## The effect of activity level and diet on the incidence of low IG readings

There was a significantly lower proportion of low IG readings ($P$ < 0.001) in dogs during periods of high activity level 5/119 (4.2%) in comparison to dogs during periods of moderate 62/119 (52.1%) and low 52/119 (43.7%) activity levels. The proportion of low IG readings at the high activity level relative to the low and moderate activity levels did not differ between the diets ($P$ = 0.544); however, within each level of activity, the proportion of low IG readings was

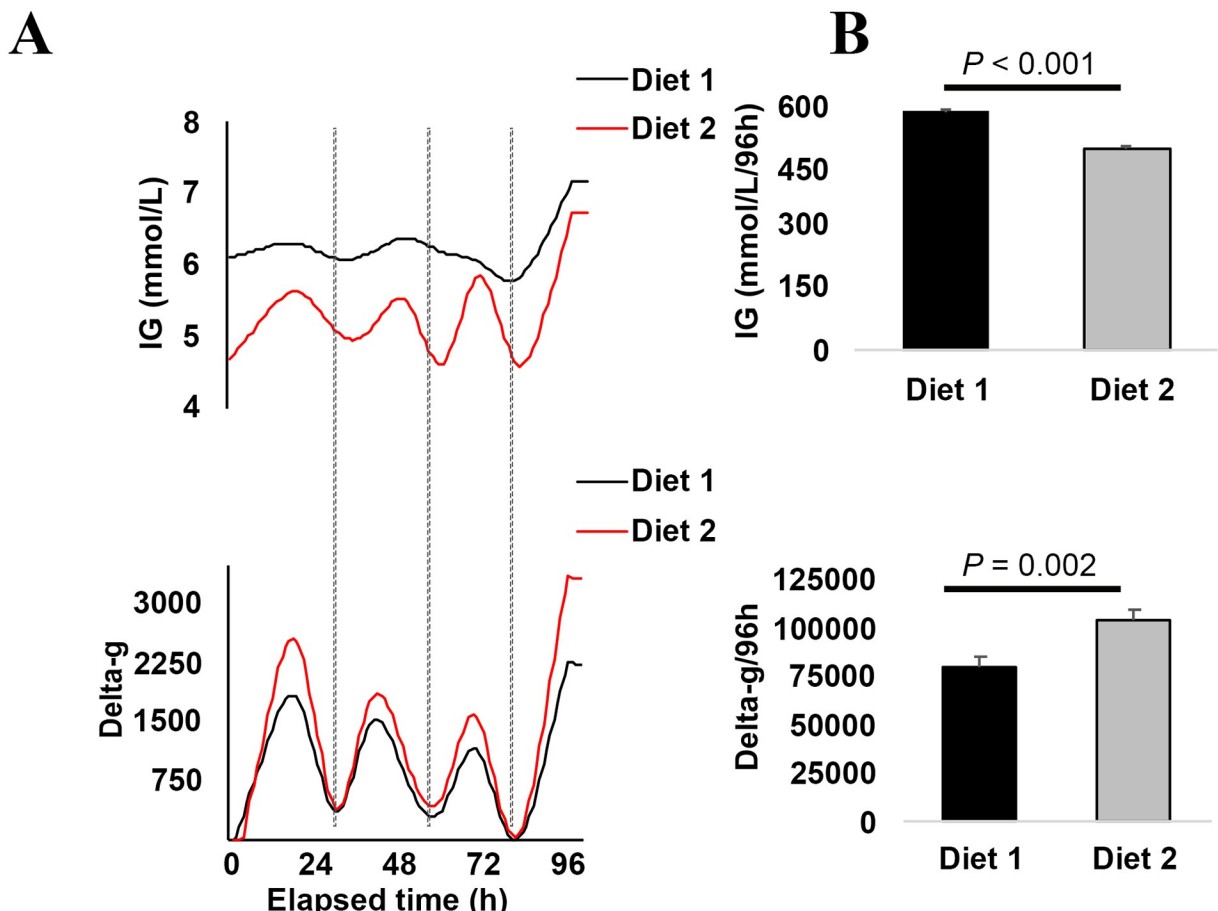

**Fig 2. The predicted IG concentration and activity level for dogs stratified by two levels of diet. 1A**. Areas under the curves. **1B**. Bar chart graphic depiction of the areas under the curves in Fig 1A. **Diet 1**, high %ME CHO low %ME fat diet; **Diet 2**, low %ME CHO high %ME fat diet; **h**, hours; **IG**, interstitial glucose. The vertical broken lines demonstrate that the lowest peaks in IG concentrations temporally lagged after the lowest peaks of delta-g.

higher in the group of dogs of Diet 2 relative to the group of dogs of Diet 1 [high activity level Diet 2 vs. Diet 1, 4 (80%) vs. 1 (20%), $P = 0.386$ moderate activity level Diet 2 vs. Diet 1, 48 (92%) vs. 4 (8%), $P < 0.001$; low activity level Diet 2 vs. Diet 1, 58 (94%) vs. 4 (6%), $P < 0.001$].

## Glucose, insulin and HOMA-IR

Mean (±SD) serum fasted glucose concentrations at the beginning and the end of the 96 h periods were 4.83 ± 0.43 and 4.66 ± 0.57 mmol/L, respectively. Median (IQR) serum insulin and HOMA-IR at the beginning of the 96 h periods were 4 mU/L (1 mU/L) and 0.80 (0.29), and at the end of the 96 h periods were 4 mU/L (2 mU/L) and 0.84 (0.44). Serum glucose, insulin, and the HOMA-IR did not differ between the beginning and the end of the 96 h periods ($P = 0.320$; $P = 0.489$; $P = 0.724$; respectively). The median (IQR) HOMA-IR was significantly lower ($P = 0.047$) at the end of the 96 h study period for Diet 2 group 0.78 (0.25) when compared to Diet 1 group 1.04 (0.56); whereas, the median (IQR) HOMA-IR did not differ between diets at the beginning of the 96 h study period (Diet 2 0.67 (0.38) vs. Diet 1 0.90 (0.78), $P = 0.061$). None of the dogs had evidence of hyperinsulinemic hypoglycemia neither at the beginning nor the end of the 96 h study periods.

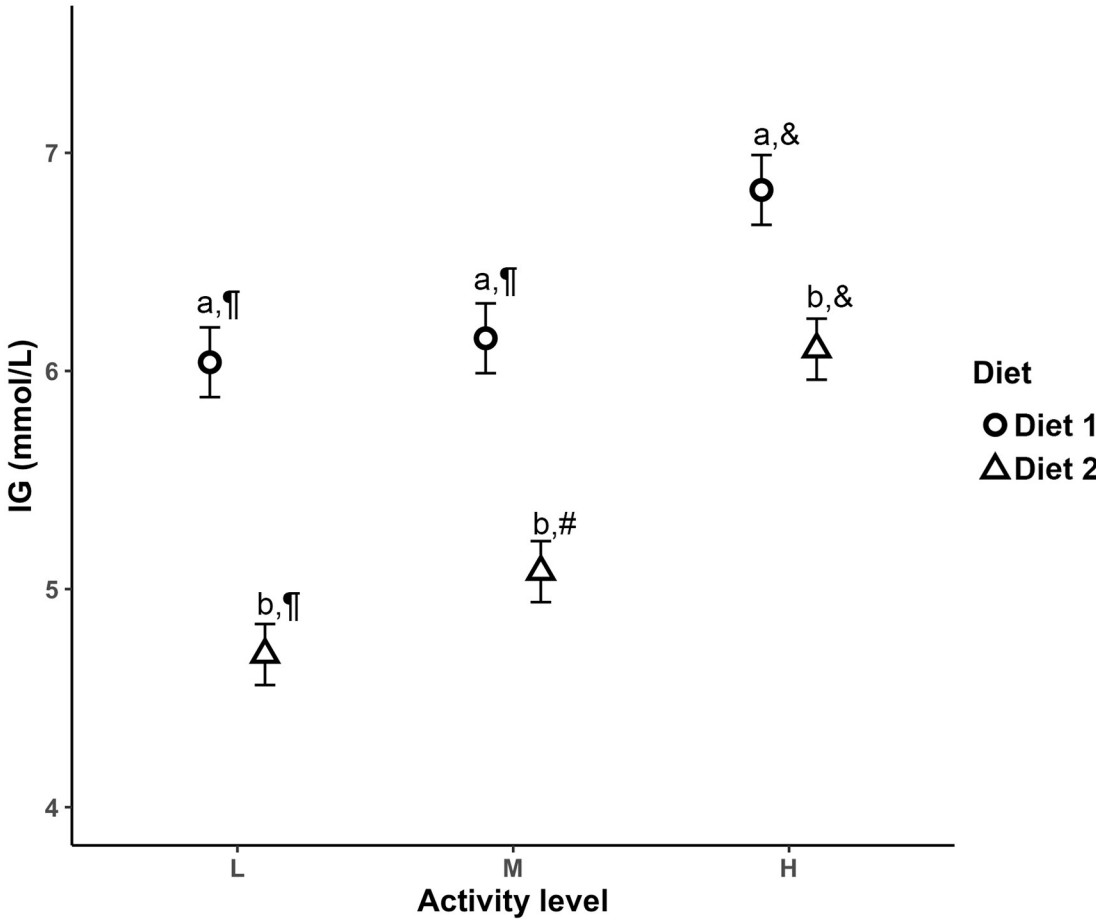

**Fig 3. Least squares means (±SE) of IG concentration derived from a flash glucose monitor as a function of three activity levels derived from triaxial accelerometer readings and stratified by two levels of diets. L**, low level of activity; **M**, moderate level of activity; **H**, high level of activity; **Diet 1**, high %ME CHO low %ME fat diet; **Diet 2**, low %ME CHO high %ME fat diet; **IG**, interstitial glucose; different letters represent a significant difference between diets per activity level; different symbols represent a significant difference between activity levels per diet; significant differences were declared at *P* < 0.050.

## Discussion

In this study we found that the adaptation to Diet 2 (high fat and ultra-low CHO with 63% and 1% ME from fat and CHO sources;) resulted in significantly lower IG and higher activity AUCs in New Zealand farm dogs compared to a diet with high CHO and low fat (Diet 1; Fig 2). We did not measure the concentrations of ketone bodies in the blood and urine of these dogs; hence we could not confirm if ketones were increased in these dogs during the 96 h periods. Based on previous studies in dogs fed similar %ME from CHO and fat sources, it is probable that following fat adaptation, ketone production and utilization was increased in these dogs, which could explain their decreased IG concentration and increased activity. In support of that, Beagle dog bitches fed a ultra-low CHO high fat diet (0% and 74% ME from CHO and fat sources) had elevated serum ketones, decreased alanine, and glucose was lower by approximately 0.83–1.1 mmol/L (15–20 mg/dL) during the week before whelping (when negative energy balance develops) compared to matched controls fed a high CHO low fat diet (44% and 30% ME from CHO and fat sources) [24]. Similar findings have been demonstrated in rodents and humans. For example, rats on a high fat diet ran 32% more than control rats that were fed

a diet with lower fat and higher CHO [25]. Adaptation to high fat diets in well-conditioned humans increases the rate of fat oxidation [26]. Ultramarathon and iron-man athletes on a high fat diet had 2.3-fold higher peak fat oxidation, and 59% higher mean fat oxidation during submaximal exercise relative to matched control ultramarathon and iron-man athletes on high CHO diets. The peak fat oxidation occurred at a higher percentage of $VO^2max$ (70.3 ± 6.3 vs. 54.9 ± 7.8). Nevertheless, these athletes did not have differences in resting and post-exercise levels of muscle glycogen demonstrating the muscle adaptation to oxidative fat metabolism [26]. As we suggested above, feeding dogs a diet of ultra-low %ME CHO and high %ME fat leads to adaptations that offer significant advantages to dogs that perform endurance work, such as New Zealand farm dogs. For example, exhaustive exercise in sled dogs that consumed a diet with 0% of metabolizable energy from a CHO source and high %ME from fat had higher albumin, total calcium, and magnesium than control dogs on diets with higher %ME from a CHO source [27]. The adaptation to high fat diets has been postulated to be beneficial because the higher albumin, total calcium, and magnesium protects against exercise-induced hypovolemia and the adaptation to high rates of muscle fat oxidation protects against muscle injury [27].

The results of this study may not directly translate to other kinds of working dogs. For example, it is unlikely that racing dogs (e.g., Greyhounds) would benefit from ultra-low CHO high fat diet as their race-day performance relies on high levels of muscle glycogen for short anerobic bursts of high-speed running. Similarly, personal assistance dogs (e.g., seeing eye dogs) that are not expected to engage in high endurance activities might not benefit from the advantages of fat adaptation like sled, military/police, search and rescue, and New Zealand farm dogs would. Maximizing the dietary benefits for each of the different fields of activity that dogs engage in our lives would require specific research tailored to the requirements of the individual field.

We also found a temporal relationship between delta-g and IG in which the direction of changes in IG concentrations followed and lagged after similar changes in delta-g (Figs 2 and 3). Mechanistically, the temporal relationship between changes in delta-g and IG could be explained by increased sympathetic tone, increased secretion of glucagon, and inhibition of insulin secretion during a high activity level which would increase IG [28]. Conversely, decreased activity levels lead to decreased sympathetic tone combined with increased insulin release, 5' adenosine monophosphate (AMP) and 5' adenosine monophosphate-activated protein kinase (AMPK)-mediated hepatic gluconeogenesis and translocation of insulin-independent glucose transporter type 4 (GLUT4) from the sarcoplasmic reticulum to the striated muscle membrane which would decrease IG [28, 29]. However, further studies are required to provide mechanistic insight to determine whether either or both of the above explanations, or alternative explanations underlie the temporal relationship between delta-g and IG as our study was not designed to investigate it.

During the 96 h of data recording there were neither low peaks of IG that followed high peaks of delta-g, nor low peaks of delta-g followed low peaks of IG. One possible explanation could be that the 96 h study periods of intense monitoring might have been too short to capture these changes. Our alternative explanation is that the dogs included in this study were physically well conditioned, had a low body fat mass, and had very high sensitivity to insulin coupled with high efficiency of energy utilization. In support of this alternative, conditioned sled dogs had higher GLUT4 per g of protein in peripheral blood mononuclear cells, lower HOMA-IR, and lower fasting concentrations of insulin and glucose than sedentary dogs. The HOMA-IR index represents the degree of sensitivity to insulin and has been validated in dogs [19, 20]. The median HOMA-IR of the dogs in this study (0.82–0.85) is similar to that reported in conditioned sled dogs [19] and provides some support to our alternative explanation.

In this study we demonstrated multiple episodes of low IG in healthy dogs. The dogs in this study had their serum insulin and glucose concentrations measured at the beginning and the end of the 96 h study periods, and none of the dogs had results that were suggestive of hyperinsulinemic hypoglycemia. Therefore, healthy dogs could have a low incidence of low IG and the frequency of low IG may be greater in dogs fed ultra-low CHO high fat diet (Diet 2). Whether episodes of low IG occur in all dogs in all activities or is only present following periods of heavy exercise remains to be determined. Also, the clinical significance of low IG is obfuscated by the absence of well-established reference intervals for IG in dogs. One possibility is that our extrapolated reference intervals for IG led to misinterpretation of the lower IG concentrations as being abnormal, and that it is normal for dogs to experience those IG concentrations, especially following exercise. Alternatively, it could be that the dogs in this study truly had physiologically low levels of IG (glucopenia) and that either clinical signs did not develop, or that the farmers did not detect the clinical signs because most occurred during the 'resting period' when the dogs were confined in the kennels and physically away from the farmers. As the study was not designed to answer this question, we cannot provide further insights. Nevertheless, the higher incidence of low IG (per our extrapolated reference interval) in dogs fed ultra-low CHO high fat diet (Diet 2) could suggest that this diet either provided insufficient glucose for hepatic output overnight, or that the adaptation to higher dietary fat reduced tissues' dependency on glucose (e.g., brain), and prevented deleterious effects of glucopenia. At this point we postulate that it is most likely the result of adaptation to higher dietary fat as discussed above.

As stated, most events of low IG occurred during the 'resting period' (84.9% vs. 15.1%). It is hard to contrast our finding with those of others because flash glucose monitoring is rarely performed on healthy individuals, and because well-established reference intervals for IG are absent. The risk for the development of hypoglycemia in diabetic humans is highest at night during the sleeping period, as sleep impairs counterregulatory-hormone responses to hypoglycemia [30–33]. In one study, hypoglycemia was induced in healthy human volunteers during early and late sleep [30]. During late sleep, there was a significantly diminished counterregulatory hormonal response (epinephrine, norepinephrine, ACTH, cortisol, and growth hormone) to the induced hypoglycemia [30]. In another study, type-1 human diabetic patients and matched healthy controls were exposed to hyperinsulinemic hypoglycemic clamps during the day, at night when they were asleep, and at night when they were awake [32]. Both, type-1 human diabetic patients and their matched healthy controls had decreased sympathoadrenal responses to hypoglycemia during sleep, but not at night if they were awake. In another study, serum glucagon concentrations were compared between human patients with type-1 diabetes and matched healthy controls [31]. Strikingly, the authors of that study demonstrated that glucagon secretion during the sleeping period was independent of glucose and insulin. Serum glucagon did not differ between type-1 diabetic patients and their matched controls, despite substantial differences in their serum glucose and insulin concentrations [31]. In a similar study, serum glucagon did not differ between human type-1 diabetics and matched healthy controls when hypoglycemia was induced during the sleeping period [33]. The nocturnal hypoglycemia phenomenon has not been previously reported in dogs. In a recent study involving 14 diabetic dogs fed a variety of commercial diets periods of low IG were documented mostly during the day [34]. The sleep patterns between dogs and humans are quite different (dogs often waking multiple times per night, human usually having a single wake/sleep cycle), which could account for inter-species differences in nocturnal hormone patterns. Alternatively, the marked difference between diets in the aforementioned study (commercial diets predominantly low in fat and with moderate to high CHO and fiber vs. ultra-low CHO high fat in our study) combined with marked differences in physical conditioning (diabetic dogs vs.

highly athletic farm working dogs), twice vs. once a day feeding, and timing of insulin administration is the essence of the difference in the timing of the low IG (as the majority of low IG reading in the Diet 1 group (high CHO low fat) in our study had also happened during the day).

This study had a few limitations. Firstly, the diets differed in respect to their micronutrient composition. However, both diets were formulated to be within the limits established by the Association of American Feed Control Officials (AAFCO), and within those limits, we are not aware of micronutrient compositions that would differ enough to affect glucose regulation. The diets would also be expected to differ in respects to digestibility and intestinal transit time, in their effects on the microflora and bacteria-derived metabolites, feelings of satiety, and potentially other processes. However, the authors posit that the paramount difference that is responsible for the differences in IG concentrations seen in these dogs is the difference in the macronutrient composition, predominantly in CHO and fat composition. A future study could reduce nutritional confounders by formulating the diets using varying proportions of the same ingredients, such that only the macronutrient proportions differ. A second limitation is that we were unable to separate a possible concurrent postprandial effect from that of resting/sleeping (between 19:00–06:00) on IG, as the dogs in this study were fed in the evening after work. Hence, the higher overall incidence of low IG between 19:00–06:00 might in part be related to a postprandial effect in addition to the inherent decreased autonomic function during the sleeping period as explained above. The length of the postprandial period in dogs has not been well characterized but in one study it was determined to be at least 6 hours [35]. In that study, dietary macronutrients composition (i.e., high CHO low fat vs. low CHO high fat) in dogs fed once daily did not significantly affect the AUC of glucose and insulin during the 6 h postprandial period, possibly implying that this is also the case in our study. However, direct comparison between that study and ours is difficult because the macronutrient compositions of diets between the two studies differed. The last limitation in the study was the unbalanced distribution of IG reading between the Diet 2 group (2306) and the Diet 1 group (1504). We contend that it is unlikely to have affected our interpretation of the results because the distribution of IG readings for each diet was relatively uniform across the 24 h and because the statistical analysis was based on a large number of observations in each group.

## Conclusion

Feeding ultra-low CHO high fat diet (Diet 2) to very active dogs was associated with decreased IG and increased activity. Interstitial glucose concentrations were positively associated with the level of activity of dogs independently of the diet. Events of low IG occurred at an incidence of 3.12% per 96 h and were predominantly seen in dogs fed ultra-low CHO high fat diet (Diet 2), occurring mostly between 19:00–06:00 following the day's work. Further work is necessary to determine if ultra-low CHO high fat diet could significantly affect work performances in New Zealand farm working dogs.

## Supporting information

**S1 Fig. Distribution of low activity quartiles from three 96-hour periods of 22 New Zealand working dogs.** The period between the two broken red vertical lines denotes the time the dogs were awake and working. Diet 1, high %ME CHO low %ME fat diet; Diet 2, ultra-low %ME CHO high %ME fat diet.
(TIF)

**S2 Fig. Distribution of IG and activity over the elapsed 96-hour periods. Diet 1**, high %ME CHO low %ME fat diet; **Diet 2**, ultra-low %ME CHO high %ME fat diet.
(TIF)

**S3 Fig. Distribution of interstitial glucose readings of 22 New Zealand working dogs over three 96-hour periods. Diet 1**, high %ME CHO low %ME fat diet; **Diet 2**, ultra-low %ME CHO high %ME fat diet; **IG**, interstitial glucose; **L-IG**, low interstitial glucose ($<$3.5 mmol/L); **N-IG**, normal interstitial glucose (3.5 mmol– 5.6 mmol/L). The horizontal dashed blue line separates the L-IG (below) from the N-IG (above). The vertical broken black lines denote the period between 19:00–06:00.
(TIF)

**S4 Fig. Distribution of low IG readings ($<$3.5 mmol/L) of 22 New Zealand working dogs over three 96-hour periods. Diet 1**, high %ME CHO low %ME fat diet; **Diet 2**, ultra-low %ME CHO high %ME fat diet. The broken red lines denote the period between 19:00–06:00.
(TIF)

**S1 Table. Nutritional analysis of trial diets.**
(DOCX)

**S2 Table. Ingredients of trial diets.**
(DOCX)

## Author Contributions

**Conceptualization:** Arnon Gal, Nick Cave, Nicolas Lopez-Villalobos, Aaron Herndon, Juila Giles, Richard Burchell.

**Data curation:** Arnon Gal, Williams Cuttance, Richard Burchell.

**Formal analysis:** Arnon Gal, Nick Cave, Nicolas Lopez-Villalobos, Aaron Herndon, Richard Burchell.

**Funding acquisition:** Arnon Gal, Juila Giles, Richard Burchell.

**Investigation:** Arnon Gal, Williams Cuttance, Aaron Herndon, Richard Burchell.

**Methodology:** Arnon Gal, Williams Cuttance, Nick Cave, Nicolas Lopez-Villalobos, Richard Burchell.

**Project administration:** Arnon Gal, Williams Cuttance.

**Resources:** Arnon Gal.

**Supervision:** Arnon Gal.

**Writing – original draft:** Arnon Gal.

**Writing – review & editing:** Arnon Gal, Williams Cuttance, Nick Cave, Nicolas Lopez-Villalobos, Aaron Herndon, Juila Giles, Richard Burchell.

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
