## [Decision Letter · Decision Letter 0]

14 Jun 2021

PONE-D-21-02404

Less is more? Ultra-low carbohydrate diet and working dogs’ performance

PLOS ONE

Dear Dr. Gal,

Thank you for submitting your manuscript to PLOS ONE. After careful consideration, we feel that it has merit but does not fully meet PLOS ONE’s publication criteria as it currently stands. Therefore, we invite you to submit a revised version of the manuscript that addresses the points raised during the review process.

We look forward to receiving your revised manuscript.

Kind regards,

Balamuralikrishnan Balasubramanian

Academic Editor

PLOS ONE

Journal Requirements:

2. In your Methods section, please provide additional details regarding participant consent from the owners of the animals. In the ethics statement in the Methods and online submission information, please ensure that you have specified (a) whether consent was informed and (b) what type you obtained (for instance, written or verbal). If the need for consent was waived by the ethics committee, please include this information.

We note that one or more of the authors are employed by a commercial company: "VetEnt Te Kuiti, Pet Doctors and North Coast Veterinary Specialist and Referral Centre"

Reviewers' comments:

Reviewer's Responses to Questions

**Comments to the Author**

1. Is the manuscript technically sound, and do the data support the conclusions?

Reviewer #1: Partly

2. Has the statistical analysis been performed appropriately and rigorously? 

Reviewer #1: Yes

3. Have the authors made all data underlying the findings in their manuscript fully available?

Reviewer #1: Yes

4. Is the manuscript presented in an intelligible fashion and written in standard English?

Reviewer #1: Yes

5. Review Comments to the Author

Reviewer #1: First, I would like to thank the author for their interesting manuscript. The work described by the authors sheds more light on the effects of macronutrient composition in dogs performing strenuous exercise. I have a few general comments for the authors, followed by some specific remarks.

- The manuscript is quite long, and some paragraphs have a lot of repetition in them. I think the readability of the manuscript may be improved by optimizing conciseness throughout the work.

- Please double check the consistency of your abbreviation use throughout the manuscript. After introducing an abbreviation, you sometimes switch back to the fully written alternative.

- The discussion section may benefit from some editing. I think it is important to focus foremost on the primary goals of this study, i.e. to determine whether IG concentrations and physical activity differ between diets, before focusing on your secondary findings, such as the moments of low IG that you found. You did this well in the abstract, so I would consider using the same order in your discussion.

- Additionally, there is a lot of focus on the cause of your findings and comparisons with previous studies, but it is also important to emphasize the consequences of your findings: what do these findings mean for working dogs? Do we all need to change their diet, or do we need to perform further specific studies? Can we extrapolate these results to all other working dogs?

- The conclusions section may also include some emphasis on why this work was important for the population you researched.

Abstract

- I would suggest changing the order of the abstract, by first stating the goals of the study, before stating your hypothesis.

25- ‘.. their body glucose levels that in turn..’Consider: their body glucose levels which in turn

25- ‘a negative effect’

Could you specify this? Maybe consider: which in turn will reduce physical activity during work

33- I think some editing may be required, consider: ‘Following the acclimation period, we continuously monitored IG concentrations with flash glucose monitoring devices, and activity using triaxial accelerometers for 96 h.

34- Dogs fed the Diet 2

I would rephrase as: dogs fed Diet 2

37- ‘Interstitial glucose’

Consider replacing with IG

39- Please specify your proportions: 119/3810 time points?

42- ‘.. was associated with increased activity, and decreased IG.’

Consider: was associated with increased activity and decreased IG concentrations.

42- ‘Interstitial glucose’ -> IG

43- However, you did not check for specific macronutrient, but only for the two diets that you used. I would consider changing this to: independently of the two diets.

Introduction

53-56 – Do you know why these diets are commonly chosen? Would you be able to give some more background for this?

65-68 – ‘Continuous absorption of glucose…’ But this does not solely depend on dietary CHO content, as the gastrointestinal passage time and interactions with other nutrient have a great effect on this.

72-74 I would suggest changing the order: This study hypothesized that dogs fed an ultra %ME CHO diet will have lower body glucose levels, when compared to dogs fed a high %ME CHO diet’, to emphasize on the ultra-low diet.

Methods

86 – did you use any block randomization to ensure that all farms had an equal part of dogs in either diet 1 or 2?

87: I would omit ‘(see description below)’

I would suggest changing the order in sentence 86-92: ‘Three 96- hour study periods took place during times of peak seasonal work activity, during which the dogs in each farm worked together as a team.’ I would include the dietary acclimation in the ‘diet’-section.

92- ‘body weight’

suggestion: bodyweight

94-96 is there any information on the validation of both methods available?

96-97 – Serum glucose and insulin…

You repeat this information in the part on the insulin and glucose assays, I would recommend omitting this information in line 96-97.

99 – this information is already stated before, I would delete one of the passages. You could, for instance, add the information on the tables earlier to line 86.

‘We recruited 22 dogs from four farms in the ….. randomizer.org) (Table 1 and 2).

This way, you can avoid this information.

101- Were any blood examinations, such as biochemistry or hematology, performed to assess for subclinical disease?

106- You could delete this sentence, as you state this information more extensively below.

116-122: I would state this in the discussion section, and not here. It would have been interesting to see whether the energy intake required to maintain bodyweight is different between both diets, but this would be a subject for further study.

141-173 Suggestion: you could consider combining some paragraphs here, to avoid switching between insulin and glucose, activity levels and back to glucose and the HOMA-IR. You could combine the information of the assays, the IG glucose measurements and the HOMA-IR, and describe the triaxial accelerometry after this.

175- Please provide the full name of HOMA-IR also in the text before using the abbreviation.

192- How did you check for normal distribution? Could you provide any information about whether the assumptions of all tests where met?

Was any power analysis performed for this study?

Results:

222-223 You also state this in your methods section, and could consider omitting it here.

The results section itself may benefit from some editing to improve conciseness. For instance, you could consider removing sentence 282, as you state this in the section heading.

It is important to consider differences between the farms: did you consider including the factor ‘farm’ in your models?

Discussion:

I think it is also beneficial for this report to consider the effects of the other macronutrients of the diets, which also differed widely. The diets also differed widely in dietary fat and protein content, two factors which have also been found to affect glucose tolerance and insulin resistance. It is therefore a possibility that the effects that you found are not due to a lack of carbohydrate content, but maybe an increase in dietary fat content.

Also, differences in housing/environment of the dogs on the different farms may have also affected your results. As an example: maybe the dogs of one of the farms were housed in colder conditions, and had a higher energy requirement (and where therefore more likely to be hypoglycemic?)

Tables and figures:

- Table 1: I think this table may be improved by swapping the ‘Farm’ and ‘variable’ columns. Additionally, I would suggest splitting these tables, as the last two rows do not fit the table headings.

- In table S1, the nutritional analysis of diet 2 is missing. Also, is there any information about the specific nutrient values of both diets (for instance, omega-3 fatty acid concentrations, vitamins and minerals)? I think it is important to provide this information, considering omega-3 fatty acids and omega 6:3 ratios have been found to alter insulin sensitivity, and therefore may affect your results.

Thank you in advance for your response.

6. PLOS authors have the option to publish the peer review history of their article (what does this mean?). If published, this will include your full peer review and any attached files.

Reviewer #1: **Yes: **N. R. Blees

---

## [Author Response · Author response to Decision Letter 0]

10 Aug 2021

The authors thank the Editor and the Reviewer for the time they invested in reviewing the manuscript and providing their valuable comments and suggestions for improving the manuscript and making it more impactful. Below we provide a point-by-point responses to each of the comments made by the Editor and Reviewer. 

Journal Requirements:

AUTHORS’ response: Files were named according to journal’s requirements.

2. In your Methods section, please provide additional details regarding participant consent from the owners of the animals. In the ethics statement in the Methods and online submission information, please ensure that you have specified (a) whether consent was informed and (b) what type you obtained (for instance, written or verbal). If the need for consent was waived by the ethics committee, please include this information.

AUTHORS’ response: We added the following sentence in the methods section (L86-87): “Dogs were recruited after receipt of owner’s verbal consent to participate in the study.”

We note that one or more of the authors are employed by a commercial company: "VetEnt Te Kuiti, Pet Doctors and North Coast Veterinary Specialist and Referral Centre"

AUTHORS’ response: Dr. Burchell was employed at Massey University during the time of the study. He left after the study was completed and has been working in a private referral center. His current employment had no impact on the study whatsoever. Like many private practitioners involved in research his involvement in the paper subsequent to leaving the university sector has been in his own time. Drs. Giles and Cuttance were not employed at Massey University at the time of the study. Their role in the study is articulated in the Authors’ Contributions sections and they did not have any conflict of interests with regards to the aforementioned study.

We added the requested sentence in L441-445 and in the online submission system.

AUTHORS’ response: We did not find anywhere in PLOS ONE Editorial system during resubmission a place where we could amend the Competing Interests Statement. In the ‘manuscript data tab’ under ‘Funding information’ there was no area to amend what the editor requested. The roles of each contributor accurately reflect their involvement at different aspects that were listed.

Reviewers' comments:

Reviewer #1: First, I would like to thank the author for their interesting manuscript. The work described by the authors sheds more light on the effects of macronutrient composition in dogs performing strenuous exercise. I have a few general comments for the authors, followed by some specific remarks.

- The manuscript is quite long, and some paragraphs have a lot of repetition in them. I think the readability of the manuscript may be improved by optimizing conciseness throughout the work.

AUTHORS’ response: We followed the reviewer suggestions and trimmed the parts in which there was unnecessary repetition.

- Please double check the consistency of your abbreviation use throughout the manuscript. After introducing an abbreviation, you sometimes switch back to the fully written alternative.

AUTHORS’ response: We made the suggested change.

- The discussion section may benefit from some editing. I think it is important to focus foremost on the primary goals of this study, i.e. to determine whether IG concentrations and physical activity differ between diets, before focusing on your secondary findings, such as the moments of low IG that you found. You did this well in the abstract, so I would consider using the same order in your discussion.

AUTHORS’ response: We restructured the discussion similar to the sequence in the abstract, as suggested by the reviewer.

- Additionally, there is a lot of focus on the cause of your findings and comparisons with previous studies, but it is also important to emphasize the consequences of your findings: what do these findings mean for working dogs? Do we all need to change their diet, or do we need to perform further specific studies? Can we extrapolate these results to all other working dogs?

AUTHORS’ response: We followed the reviewer suggestions (see below).

For the question “what do these findings mean for working dogs?”

L321-323: “As we suggested above, feeding dogs a diet of ultra-low %ME CHO and high %ME fat leads to adaptations that offer significant advantages to dogs that perform endurance work, such as New Zealand farm dogs”.

L327-329: “The adaptation to high fat diets has been postulated to be beneficial because the higher albumin, total calcium, and magnesium protects against exercise-induced hypovolemia and the adaptation to high rates of muscle fat oxidation protects against muscle injury”.

Do we all need to change their diet, or do we need to perform further specific studies? Can we extrapolate these results to all other working dogs?

L330-337: “The results of this study may not directly translate to other kinds of working dogs. For example, it is unlikely that racing dogs (e.g., Greyhounds) would benefit from ultra-low CHO high fat diet as their race-day performance relies on high levels of muscle glycogen for short anerobic bursts of high-speed running. Similarly, personal assistance dogs (e.g., seeing eye dogs) that are not expected to engage in high endurance activities might not necessary benefit from the advantages of fat adaptation like sled, military/police, search and rescue, and New Zealand farm dogs would. Maximizing the dietary benefits for each of the different fields of activity that dogs engage in our lives would require specific research tailored to the requirements of the individual field”.

- The conclusions section may also include some emphasis on why this work was important for the population you researched.

AUTHORS’ response: Identifying whether the increased activity that we found is translated to increased work performances in the field should really be the next step before making a recommendation to change the diet for all working dogs.

We added the following sentence (L438-439): “Further work is necessary to determine if ultra-low CHO high fat diet could significantly enhance work performances in New Zealand farm working dogs”.

Abstract

- I would suggest changing the order of the abstract, by first stating the goals of the study, before stating your hypothesis.

AUTHORS’ response: We made the suggested change (L24-28).

25- ‘.. their body glucose levels that in turn..’Consider: their body glucose levels which in turn

AUTHORS’ response: The sentence was changed as follow: “We hypothesized that feeding working farm dogs an ultra-low CHO diet would reduce their IG concentrations which in turn would reduce physical activity during work” (L27-28).

25- ‘a negative effect’

Could you specify this? Maybe consider: which in turn will reduce physical activity during work

AUTHORS’ response: We made the suggested change (L28).

33- I think some editing may be required, consider: ‘Following the acclimation period, we continuously monitored IG concentrations with flash glucose monitoring devices, and activity using triaxial accelerometers for 96 h.

AUTHORS’ response: We made the suggested change (L32-34).

34- Dogs fed the Diet 2

I would rephrase as: dogs fed Diet 2

AUTHORS’ response: We made the suggested change (L34).

37- ‘Interstitial glucose’

Consider replacing with IG

AUTHORS’ response: We made the suggested change (L37).

39- Please specify your proportions: 119/3810 time points?

AUTHORS’ response: To explain what the proportion relates to we added the word ‘readings’ (L39): “The overall incidence of low IG readings (< 3.5 mmol/L) was 119/3810 (3.12%)…”

42- ‘.. was associated with increased activity, and decreased IG.’

Consider: was associated with increased activity and decreased IG concentrations.

AUTHORS’ response: We changed the sentence as follow: “We conclude that feeding Diet 2 (ultra-low CHO high fat diet) to working farm dogs was associated with increased activity despite decreased IG concentrations” (L41-42).

42- ‘Interstitial glucose’ -> IG

AUTHORS’ response: We made the suggested change (L42).

43- However, you did not check for specific macronutrient, but only for the two diets that you used. I would consider changing this to: independently of the two diets.

AUTHORS’ response: We made the suggested change (L43).

Introduction

53-56 – Do you know why these diets are commonly chosen? Would you be able to give some more background for this?

AUTHORS’ response: The answer is a combination of practical convenience, and success of a single dominant brand. Dogs were historically fed meat and offal from livestock killed on farm (“home-kill”). When home-kill was fed alone, it constituted a very low to zero carbohydrate diet. 

For decades, the most popular commercial food was Tux® (Nestle-Purina), originally about 45% of ME as CHO, then more recently 35% ME (Tux Energy®).

During that time, farmers have varied the amount of home-kill and Tux fed, which ranged from 100% home-kill, through 50:50, to 100% Tux (Singh et al).

Singh, I., Tucker, L. A., Gendall, P., Rutherfurd-Markwick, K. J., Cline, J., & Thomas, D. G. (2011). Age, breed, sex distribution and nutrition of a population of working farm dogs in New Zealand: results of a cross-sectional study of members of the New Zealand Sheep Dog Trial Association. New Zealand Veterinary Journal, 59(3), 133-138. 10.1080/00480169.2011.567967

65-68 – ‘Continuous absorption of glucose…’ But this does not solely depend on dietary CHO content, as the gastrointestinal passage time and interactions with other nutrient have a great effect on this.

AUTHORS’ response: The reviewer is correct. What we meant to say was that higher dietary glucose would translate to an overall increased absorption of glucose from the GI tract compared with diets low in CHO. 

We changed the sentence (L65-68) to “Therefore, feeding diets with high CHO content to dogs that are undergoing strenuous and prolonged physical exercise might support the maintenance of glucose levels for a longer time through maximizing muscle and hepatic glycogen stores and glucose absorption from the gastrointestinal tract”.

72-74 I would suggest changing the order: This study hypothesized that dogs fed an ultra %ME CHO diet will have lower body glucose levels, when compared to dogs fed a high %ME CHO diet’, to emphasize on the ultra-low diet.

AUTHORS’ response: The sentence was revised as follow (72-74): “This study tested the hypothesis that dogs fed an ultra-low CHO high fat diet will have lower body glucose levels than dogs fed a high CHO low fat diet, evidenced by lower IG levels and reduced physical activity (Fig. 1)”

Methods

86 – did you use any block randomization to ensure that all farms had an equal part of dogs in either diet 1 or 2? 

AUTHORS’ response: Yes. The exact allocation of number of dog/diet/farm is evident in Table 2 under Farms. We did our best to balance the dogs between diets within farms however when you’re dealing with a small uneven number of dogs/farm it is somewhat challenging…

87: I would omit ‘(see description below)’

I would suggest changing the order in sentence 86-92: ‘Three 96- hour study periods took place during times of peak seasonal work activity, during which the dogs in each farm worked together as a team.’ I would include the dietary acclimation in the ‘diet’-section.

AUTHORS’ response: We omitted ‘(see description below)’ and changed the sentence as follow: “Acclimation of the dogs to the two diets started one month prior to the commencement of each of the three 96 h study periods, which took place during times of peak seasonal work activity, during which the dogs in each farm worked together as a team.” (L89-92)

If we had completely removed the acclimation part of the sentence, then the sentence below that explains why the study spun between Oct-Nov becomes out-of-context. We hope the reviewer finds it suitable.

92- ‘body weight’

suggestion: bodyweight

AUTHORS’ response: We made the suggested change (L95).

94-96 is there any information on the validation of both methods available?

AUTHORS’ response: Yes. 

For the Flash Glucose Monitoring system: 

• Corradini S, Pilosio B, Dondi F, Linari G, Testa S, Brugnoli F, Gianella P, Pietra M, Fracassi F. Accuracy of a Flash Glucose Monitoring System in Diabetic Dogs. J Vet Intern Med 2016;30:983-988.

• Del Baldo F, Canton C, Testa S, Swales H, Drudi I, Golinelli S, Fracassi F. Comparison between a flash glucose monitoring system and a portable blood glucose meter for monitoring dogs with diabetes mellitus. J Vet Intern Med 2020;34:2296-2305.

• Malerba E, Cattani C, Del Baldo F, Carotenuto G, Corradini S, Golinelli S, Drudi I, Fracassi F. Accuracy of a flash glucose monitoring system in dogs with diabetic ketoacidosis. J Vet Intern Med 2020;34:83-91.

• Silva DD, Cecci GRM, Biz G, Chiaro FN, Zanutto MS. Evaluation of a flash glucose monitoring system in dogs with diabetic ketoacidosis. Domest Anim Endocrinol 2021;74:106525.

For the Heyrex accelerometer:

• Albright JD, Seddighi RM, Ng Z, Sun X, Rezac DJ. Effect of environmental noise and music on dexmedetomidine-induced sedation in dogs. PeerJ 2017;5:e3659.

• Lee AH, Detweiler KB, Harper TA, Knap KE, de Godoy MRC, Swanson KS. Physical Activity Patterns of Free Living Dogs Diagnosed with Osteoarthritis. J Anim Sci 2021.

• Mejia S, Duerr FM, Salman M. Comparison of activity levels derived from two accelerometers in dogs with osteoarthritis: Implications for clinical trials. Vet J 2019;252:105355.

96-97 – Serum glucose and insulin…

You repeat this information in the part on the insulin and glucose assays, I would recommend omitting this information in line 96-97.

AUTHORS’ response: We made the suggested change.

99 – this information is already stated before, I would delete one of the passages. You could, for instance, add the information on the tables earlier to line 86.

‘We recruited 22 dogs from four farms in the ….. randomizer.org) (Table 1 and 2).

This way, you can avoid this information.

AUTHORS’ response: We made the suggested change.

“We recruited 22 dogs from four farms in the North Island of New Zealand and randomized the dogs into two balanced groups with respect to diet (n=11 each) using an online randomization tool (www.randomizer.org) (Tables 1 and 2). Dogs were recruited after receipt of each owner’s verbal consent to participate in the study. Dogs were only included in the study if they had been assessed by a physical exam by at least one of the authors and by the history provided by the owner and were deemed to be in good health” (L84-89).

101- Were any blood examinations, such as biochemistry or hematology, performed to assess for subclinical disease?

AUTHORS’ response: The short answer is no. The longer explanation is that dogs that are involved in high level of work such as these dogs cannot perform their work if they have underlying conditions and the owners quickly see that there is alteration in their performances. Also, it was not in our budget to prescreen with intensive bloodwork and additional imaging which would have been required in order to have confidence that the dogs do not have an underlying condition. We qualify our observations and wrote that the dogs were ‘deemed’ in good health and ideal body condition following physical exam and owner’s history. 

106- You could delete this sentence, as you state this information more extensively below.

AUTHORS’ response: We made the suggested change.

116-122: I would state this in the discussion section, and not here. It would have been interesting to see whether the energy intake required to maintain bodyweight is different between both diets, but this would be a subject for further study.

AUTHORS’ response: We prefer to keep these sentences in the Diets paragraph within the Methods section (unless the reviewer would strongly object) for the reason that the reader immediately understands the rationale for why we chose to do what we did.

141-173 Suggestion: you could consider combining some paragraphs here, to avoid switching between insulin and glucose, activity levels and back to glucose and the HOMA-IR. You could combine the information of the assays, the IG glucose measurements and the HOMA-IR, and describe the triaxial accelerometry after this.

AUTHORS’ response: We combined the insulin and HOMA-IR paragraphs together and kept the FGM and triaxial accelerometry in one paragraph. We think that it keeps the flow better and the structure more coherent. 

175- Please provide the full name of HOMA-IR also in the text before using the abbreviation.

AUTHORS’ response: We made the suggested change (L171)

192- How did you check for normal distribution? Could you provide any information about whether the assumptions of all tests where met?

Was any power analysis performed for this study?

AUTHORS’ response: We added the following sentence (L196-197) indicating how we assessed for normal distribution: “The data were examined for normal distribution by inspection of Q-Q plots, histogram, and by the Shapiro-Wilk test.” 

Presence or absence of normal distribution was a criterion that determined whether ANOVA (normally distributed data) or nonparametric (nonnormally distributed data) tests had been used (indicated in L207-210). 

We also added the following paragraph (L189-193): “A priori power analysis (G*Power software version 3.1.9.2) indicated that 9 dogs in each group will be sufficient to detect a difference between two dependent IG means of 100 mg/dL and 70 mg/dL in an individual dog with SD of 30 mg/dL, power of 0.8, alpha error probability of 0.05, and a correlation coefficient of 0.6 between measurements. To account for potential losses of dogs during the study period (i.e., dislodgement of the sensor), we recruited 11 dogs per group.”

The modelling of IG and activity using ASReml do not require the traditional assumptions to perform analysis of variance using the least-squares method (normality, homogenous variance, independent residual errors and additive effects), because the solutions of the model were obtained using restricted maximum likelihood procedures, which do not require the homogeneity of variance assumption; and the correlations among random residuals were accounted for in the modelling of observations for each dog using splines. 

Results:

222-223 You also state this in your methods section, and could consider omitting it here.

AUTHORS’ response: We made the suggested change.

The results section itself may benefit from some editing to improve conciseness. 

AUTHORS’ response: We trimmed a few sentences in different paragraphs of the results that were repetitive or redundant. 

For instance, you could consider removing sentence 282, as you state this in the section heading.

AUTHORS’ response: We made the suggested change.

It is important to consider differences between the farms: did you consider including the factor ‘farm’ in your models?

AUTHORS’ response: We tested for a ‘farm’ effect when we analyzed the data and since ‘farm’ did not significantly affect the model it was omitted. 

Discussion:

I think it is also beneficial for this report to consider the effects of the other macronutrients of the diets, which also differed widely. The diets also differed widely in dietary fat and protein content, two factors which have also been found to affect glucose tolerance and insulin resistance. It is therefore a possibility that the effects that you found are not due to a lack of carbohydrate content, but maybe an increase in dietary fat content.

AUTHORS’ response: The reviewer is correct. Throughout the revised manuscript we referred to Diet 2 as “ultra-low CHO high fat” and Diet 1 as “high CHO low fat” as they are linked together and decreasing either CHO or fat in the diet leads to reciprocal increase in the other component. While the diets also differed in %ME protein we do not think that this difference [23% (Diet 1) vs 37% (Diet 2)] had the same effect as the marked differences in CHO and fat. In the discussion we also ascribe the dietary effect that we found in our study to the process of fat adaptation.

Also, differences in housing/environment of the dogs on the different farms may have also affected your results. As an example: maybe the dogs of one of the farms were housed in colder conditions, and had a higher energy requirement (and where therefore more likely to be hypoglycemic?)

AUTHORS’ response: The four farms in the study were from the Waikato region in New Zealand. Housing conditions of farm dogs in New Zealand are quite standard (outdoors kennel). Therefore, we think that it is very unlikely that the dogs in the different farms were subjected to very different terrains or ambient temperatures. Also, as we knew that we could not completely control for an identical workload between farms (although we think the workloads were quite similar given we aimed the 96-hour periods to occur at peak seasonal activity) we made sure to balance the diet within the farms. Hence, despite the inherent minimal differences between farms in ambient temperature, terrain, and workload we think that the major effects on IG and activity are foremost due to the diet.

Tables and figures:

- Table 1: I think this table may be improved by swapping the ‘Farm’ and ‘variable’ columns. Additionally, I would suggest splitting these tables, as the last two rows do not fit the table headings.

AUTHORS’ response: We made the suggested change.

- In table S1, the nutritional analysis of diet 2 is missing. 

AUTHORS’ response: We added the nutritional analysis of diet 2 that was accidently deleted.

Also, is there any information about the specific nutrient values of both diets (for instance, omega-3 fatty acid concentrations, vitamins and minerals)? I think it is important to provide this information, considering omega-3 fatty acids and omega 6:3 ratios have been found to alter insulin sensitivity, and therefore may affect your results.

AUTHORS’ response: A full nutrient profile for commercial diets is $2,500 and beyond the budget of this trial, thus unfortunately, we don’t have the full nutritional analysis of each diet. That is proprietary information, and we did not have access to that, beyond what information was included in the tables for Tux (Diet 1). A little more was available for the K9 Natural product (Diet 2), but not for both diets. 

Within the scope of AAFCO requirements, we are not aware that insulin sensitivity will be affected by micronutrient concentrations in a manner that exceeds the effect from the massive difference in CHO content.

The effect of n-3 PUFA has been shown in diabetic humans. The effects of n-3 PUFA in healthy dogs are less clear. One study showed that enrichment of the diet with n-3 PUFA at 0.2% (as fed basis) had no effect on insulin sensitivity (Irvine et al 2002), and another showed no relationship between serum n-3 PUFA concentrations in dogs and serum insulin (when the dogs were eating various unreported diets), whilst obesity did have a significant effect.

The ultra-low CHO diet (Diet 1) contains a fish oil (Hoki oil), and the n-3 PUFA content is 0.28% (as fed basis), and thus is almost exactly the amount tested by Irvine et al (2002). The dry diet contains a small amount of the n-3 alpha linolenic acid, but no significant amount of long chain n-3 PUFA, and thus would be similar to the control diet in the study by Irvine et al.

We think that there is no reason to suspect that the different PUFA contents of the diets have a significant effect on insulin sensitivity. Compared with the effect of the difference in macronutrients (i.e. CHO, fat, and protein), any effect from micronutrient differences is argued to be small.

Streeter RM, Struble AM, Mann S, Nydam DV, Bauer JE, Castelhano MG, Todhunter RJ, Cummings BP, Wakshlag JJ. The associations between serum adiponectin, leptin, C-reactive protein, insulin, and serum long-chain omega-3 fatty acids in Labrador Retrievers. Vet Med (Auckl). 2015 Apr 8;6:103-110.

Irvine AJ, Butterwick R, Watson T, Millward DJ, Morgan LM. Dietary supplementation with (n-3) polyunsaturated fatty acids does not affect insulin sensitivity in healthy Labrador retriever dogs. J Nutr. 2002 Jun;132(6 Suppl 2):1709S-10S

Thank you in advance for your response.

AUTHORS’ response: Thank you for providing the feedback required to improve and better our manuscript.

---

## [Decision Letter · Decision Letter 1]

19 Oct 2021

PONE-D-21-02404R1Less is more? Ultra-low carbohydrate diet and working dogs’ performancePLOS ONE

Dear Dr. Gal,

Thank you for submitting your manuscript to PLOS ONE. After careful consideration, we feel that it has merit but does not fully meet PLOS ONE’s publication criteria as it currently stands. Therefore, we invite you to submit a revised version of the manuscript that addresses the points raised during the review process. Please submit your revised manuscript by Dec 03 2021 11:59PM. If you will need more time than this to complete your revisions, please reply to this message or contact the journal office at plosone@plos.org. Please include the following items when submitting your revised manuscript:A rebuttal letter that responds to each point raised by the academic editor and reviewer(s). You should upload this letter as a separate file labeled 'Response to Reviewers'.A marked-up copy of your manuscript that highlights changes made to the original version. You should upload this as a separate file labeled 'Revised Manuscript with Track Changes'.An unmarked version of your revised paper without tracked changes. You should upload this as a separate file labeled 'Manuscript'.If applicable, we recommend that you deposit your laboratory protocols in protocols.io to enhance the reproducibility of your results. Protocols.io assigns your protocol its own identifier (DOI) so that it can be cited independently in the future. For instructions see: https://journals.plos.org/plosone/s/submission-guidelines#loc-laboratory-protocols. Additionally, PLOS ONE offers an option for publishing peer-reviewed Lab Protocol articles, which describe protocols hosted on protocols.io. Read more information on sharing protocols at https://plos.org/protocols?utm_medium=editorial-email&utm_source=authorletters&utm_campaign=protocols.

We look forward to receiving your revised manuscript.

Kind regards,

Balamuralikrishnan Balasubramanian

Academic Editor

PLOS ONE

Journal Requirements:

Additional Editor Comments (if provided):

Amend the Reviewer 2 Comments.

Reviewers' comments:

Reviewer's Responses to Questions

**Comments to the Author**

1. If the authors have adequately addressed your comments raised in a previous round of review and you feel that this manuscript is now acceptable for publication, you may indicate that here to bypass the “Comments to the Author” section, enter your conflict of interest statement in the “Confidential to Editor” section, and submit your "Accept" recommendation.

Reviewer #1: All comments have been addressed

Reviewer #2: (No Response)

2. Is the manuscript technically sound, and do the data support the conclusions?

Reviewer #1: (No Response)

Reviewer #2: Partly

3. Has the statistical analysis been performed appropriately and rigorously? 

Reviewer #1: (No Response)

Reviewer #2: No

4. Have the authors made all data underlying the findings in their manuscript fully available?

Reviewer #1: (No Response)

Reviewer #2: Yes

5. Is the manuscript presented in an intelligible fashion and written in standard English?

Reviewer #1: (No Response)

Reviewer #2: Yes

6. Review Comments to the Author

Reviewer #1: All comments are thoroughly addressed, thank you to the authors for their revised version of the manuscript.

Reviewer #2: Overall, a very interesting manuscript. Additional work investigating the effect of nutrition on active, working dogs is much needed. My biggest concern is about statistical analysis of the data, which is outlined in the methods section below.

Abstract:

L29 – Sentence reads awkwardly. The dogs were randomized to a diet then fed for a month, but the sentence almost reads as if the randomization took a month.

L31 – Remove additional parenthesis around “Diet 1”

Introduction:

General comment – Introduction would benefit from an additional paragraph on why a low CHO diet is selected. The second paragraph of the introduction describes a need for CHO during intense exercise, but then why are low CHO diets used at all? In the discussion, the focus shifts to how low CHO, high fat diets may be advantageous, but this should be introduced in introduction.

L55 – Remove extra space in “(%ME )”

L56 – Suggest expanding on why this is an important area of study (it is more than it being unknown)

L72-76 – Suggest switching the hypothesis and study aims sentences to first describe study aims then hypothesis/expected outcome

Materials and Methods:

L85 – Please give additional information regarding how groups were randomized. Was the randomization balanced across farms, sex, and breed? It does not appear so based on Table 2. It may be simply due to the dogs available at each farm, but please state that here if that is the case.

L86 – Italicize “n”

L88 - Within what time period were the dogs assessed? Within weeks? A year?

L90 – Remove extra space between “which” and “took”

L94 – Suggest removing first person here and throughout - “Bodyweight was recorded at recruitment, then at the beginning and end of each of the 96 h periods.”

L97 – When was the accelerometer fitted? At recruitment? Was there a baseline activity collection period?

L126 – Serum tubes allowed to clot before what? Assuming this means allowed to clot before being put on ice since the samples were not processed on site, but please clarify.

L136 – Remove comma after “whereas”

L149-150 – Why no baseline measurement during month of acclimation?

L159-160 – Please include any relevant citations for established blood glucose levels

L168 – Please provide inter- and intra-assay variation and sensitivity for this assay

L171 – Suggest brief explanation of what HOMA-IR is. What is normal? What is expected in this population of dogs?

L174-177 – I do not see the purpose of this arbitrary distinction into activity levels, particularly with so few animals who were all highly active working dogs. This becomes a problem later on in the statistical analysis section when activity (a response variable) is used as a fixed effect for IG analysis.

L176 – Is the =< supposed to be ≤ or <!--?<br /L198-199 – Were data transformations attempted?

L201-203 – several questions:

- Why is time not included as a fixed effect?

- Above, you mention activity and inactivity periods, but these are also not reflected in this analysis

- What about farm, sex, age, breed, etc.? If these were evaluated as fixed effects then dropped, that should still be documented here.

- Activity is a response variable, not a fixed effect. If you wanted to include “activity level” as a fixed effect, you would need baseline activity data by which to block the dogs by activity level prior to diet allocation & acclimation

L207-208 – Again, were any transformations attempted prior to analysis as a nonparametric variable?

L208 – Remove comma after “whereas”

L209 – Why GLM here but MIXED above? What was included in the GLM model?

L213 – Why not include time & diet x time interaction here?

L214 – Is “activity level” here referring to the “activity levels” described above (L174-177)? Why examine activity level as a dependent variable here but a fixed effect in the linear models above?

Results:

L221 – How was BW statistically analyzed? This is not described in methods. The description of BW effects is a bit clunky. Were diet and time fixed effects significant? The interaction? You jump straight to describing the individual diet x time interactions without presenting results of main effects.

L225 – Suggest authors use consistency with respect to P-value reporting – 2 decimals used here versus three in next line. Recommend confirming if journal has standard for reporting P-values.

L226 – With the standard errors reported here, these weights should not have a statistical difference. Something is not correct with this analysis.

L229 – What about time effect? Diet x time interaction?

L242 – Your hypothesis is that low CHO diets lead to low IG which in turn reduces activity. Now you are trying to attribute activity to IG levels! Again, activity is a response variable, not a fixed effect. Dogs’ activity is likely related to IG, but that is not what this study was designed to assess. This section should be removed.

L256 – Period of the day is not included in your statistical analysis (L201-203) as a fixed effect.

L256 – Is the incidence of low IG more relevant than time spent at low IG? How much total time was spent at low IG? The potential diet x time interaction is important.

L266 – Remove extra space between “Diet 2” and “group”

L269-270 – Would it be expected for the majority of low IG readings to occur in the activity period? Would be nice to mention whether or not this was expected.

L275 – Remove comma after “whereas”

L281 – Again, activity is not a fixed effect, and your hypothesis has the reverse – low IG reducing activity. This section should be removed.

L299 – Consider re-wording as there is no such thing as “inappropriate” serum insulin.

Discussion:

L304 – These results were presented above, the reference to the figure is not needed here.

L307 – Suggest removing “most”

L314 – Reword “people” as “humans” throughout

L321-322 – Where is it suggested that low fat/high CHO diet offers advantages? Your hypothesis is that lower CHO/high fat diet will reduce activity.

L323-326 – These two sentences are very repetitive. Suggest condensing into one sentence.

L334 – Reword to “necessarily”

L339-340 – Here you could discuss how activity could have an impact on IG and recommend additional research since your study was not designed to examine this effect.

L340 – Remove reference to figures from discussion

L347 – Suggest further discussion at the end of this paragraph as to the potential significance of this observed relationship.

L349-350 – Awkward wording, suggest re-working. Since the hypothesis is that low IG leads to reduced activity, suggest focusing on that. May discuss potential for higher activity to reduce IG as avenue for future research, but not really relevant here.

L351 – Expand on the potential limitation of the 96 h collection period. What time period would have been better?

L358 – Suggest “of this alternative”

L360-361 – Suggest removing “The incidence of low IG episodes was 3.12% per 96 h and involved 10/22 (45.5%) of the dogs” from the discussion since it has already been reported in results section.

L363 – Similar to comment in L299 above, please remove “inappropriate” and re-word.

L364 – Suggest removing “we conclude that”

L365 – Is the fact that healthy, active dogs can have low IG really a surprising conclusion/finding of this study? Suggest writing different sentence to link discussion between serum insulin/glucose and IG

L373 – What effect might feeding time/frequency have on IG?

L377 – How does the evening feeding time play into this? Suggest further discussion

L409 – Any suggestions for future investigation regarding IG monitoring? Potential future work could be worked in throughout discussion.

L418 – Suggest addition of how this limitation could be overcome, such as formulating the diets to be similar in every respect except for source of energy (CHO vs. fat)

Conclusion:

L435 – Spell out abbreviation when it begins a sentence.

Tables and Figures:

Fig. 1 – in caption, should read “an ultra-low CHO…” not “a”

Table 2 – Diet randomization is clearly not balanced across farms, particularly farm 3. If there is a reason for this, please include in methods section.

Table 3 – Recommend removal since activity should not be used as a fixed effect.

7. PLOS authors have the option to publish the peer review history of their article (what does this mean?). If published, this will include your full peer review and any attached files.

Reviewer #1: **Yes: **N. R. Blees

Reviewer #2: No

---

## [Author Response · Author response to Decision Letter 1]

1 Dec 2021

PONE-D-21-02404R: Less is more? Ultra-low carbohydrate diet and working dogs' performance 

The authors thank the Editor and both Reviewers for the time they invested in reviewing the manuscript and providing their valuable comments and suggestions for improving the manuscript and making it more impactful. Below we provide a point-by-point responses to each of the comments made by the Editor and Reviewer 2. 

REVIEWER 1

Authors’ response: Thank you!

REVIEWER 2

Reviewer #2: Overall, a very interesting manuscript. Additional work investigating the effect of nutrition on active, working dogs is much needed. My biggest concern is about statistical analysis of the data, which is outlined in the methods section below.

Authors' response: After we went through the rebuttal process, we think that the main problem that Reviewer 2 had with our statistical analysis stemmed from a problem with terminology on our end for which we apologize. Specifically, we think that the use of ‘activity’ and ‘activity level’ interchangeably confused Reviewer 2 with regards to what was used as a response variable and what was used as a fixed effect in our analyses. As we explain below, we therefore used ‘delta-g’ as a response variable (derived from the accelerometer recordings), and we stratified ‘delta-g’ to three levels (L/M/H) of activity to assess the effect of ‘activity level’, ‘diet’, and ‘time’ (across the 96-hour of data recording and split between ‘working period’ and ‘resting period’) and interactions between diet x time on IG.

Abstract:

L29 – Sentence reads awkwardly. The dogs were randomized to a diet then fed for a month, but the sentence almost reads as if the randomization took a month.

Authors' response: We changed the sentence as follows (L29): "At each farm, dogs were randomized to one of two diets and had a month of dietary acclimation to their allocated diet. "

L31 – Remove additional parenthesis around "Diet 1"

Authors' response: The parenthesis was removed.

Introduction:

General comment – Introduction would benefit from an additional paragraph on why a low CHO diet is selected. The second paragraph of the introduction describes a need for CHO during intense exercise, but then why are low CHO diets used at all? In the discussion, the focus shifts to how low CHO, high fat diets may be advantageous, but this should be introduced in introduction.

Authors' response: There are two primary feeding practices for working dogs in New Zealand. The reasons why these are the two common practices are multifactorial and involves farmers' personal beliefs and financial concerns (commercially available diets are generally more expensive than home-kill meat-based homemade diets) (L53-56). 

We added the following in L56: “Dogs do not require dietary CHO, and exercising sled dogs may have improved performance when fed a high-fat CHO-free diet [2]. However, the concept of “cross-over”, where muscle utilization of CHO for ATP production increases with increased exercise intensity, has been demonstrated in several different mammalian species, including dogs [3]. Since the intensity of activity engaged by working farm dogs in New Zealand is likely to be greater than the sustained moderate-intensity endurance activity of sled dogs, New Zealand’s working farm dogs may be unable to maintain normal body glucose levels when fed a low CHO diet.”

Reviewer 1 was already concerned about the overall length of the manuscript and we are reluctant to add additional information beyond this. 

L55 – Remove extra space in "(%ME )"

Authors' response: we removed the extra space.

L56 – Suggest expanding on why this is an important area of study (it is more than it being unknown)

Authors' response: We added the following in L56: “Dogs do not require dietary CHO, and exercising sled dogs may have improved performance when fed a high-fat CHO-free diet [2]. However, the concept of “cross-over”, where muscle utilization of CHO for ATP production increases with increased exercise intensity, has been demonstrated in several different mammalian species, including dogs [3]. Since the intensity of activity engaged by working farm dogs in New Zealand is likely to be greater than the sustained moderate-intensity endurance activity of sled dogs, New Zealand’s working farm dogs may be unable to maintain normal body glucose levels when fed a low CHO diet.”

L72-76 – Suggest switching the hypothesis and study aims sentences to first describe study aims then hypothesis/expected outcome

Authors' response: We would prefer not to do so unless the Reviewer/Editor is adamant that we change the order. Our rationale is that the aims are derived from the hypothesis and not vice versa. 

Materials and Methods:

L85 – Please give additional information regarding how groups were randomized. Was the randomization balanced across farms, sex, and breed? It does not appear so based on Table 2. It may be simply due to the dogs available at each farm, but please state that here if that is the case.

Authors' response: The randomization was based on balancing the diets across farms and dog availability and not on sex, age, breed, or bodyweight because it was challenging to find farms and farmers that were willing to participate. Hence, we were unable to be too selective. 

The sentence was changed as follows (L91): "We recruited 22 dogs from four farms on the North Island of New Zealand and randomized the dogs into two balanced groups with respect to diet (n=11 each) using an online randomization tool (www.randomizer.org); there was no attempt to balance the dogs per age, sex, breed, or bodyweight during randomization and recruitment (Tables 1 and 2)."

L86 – Italicize "n"

Authors' response: we italicized it.

L88 - Within what time period were the dogs assessed? Within weeks? A year?

Authors' response: One of the authors (WC) has been providing regular veterinary care to the farmers in this region. It is important to mention that WC gauged the interest of the farmers to participate in the study and enroll their dogs with respect to the high level of work those dogs were expected to do in the month to come. WC assessed the dogs on the farms of the farmers that indicated they were keen to participate and determined if the dogs were deemed appropriate to be enrolled in the study. Once WC decided to enroll the dogs, they were enrolled within days (i.e., the time it took us to ship the diets to the farmers in those farms). Then the acclimation period started and a month after that, the 96-hour of data recording commenced, which took place at the peak seasonal workload.

L90 – Remove extra space between "which" and "took"

Authors' response: we removed the extra space.

L94 – Suggest removing first person here and throughout - "Bodyweight was recorded at recruitment, then at the beginning and end of each of the 96 h periods."

Authors' response: We prefer to use an 'active voice' rather than a 'passive voice' as much as possible. This is a stylistic preference, and the Journal does not have a specific guideline regarding it. Unless the Reviewer is adamant about that, we prefer to leave it like that. The Editor may also have a preference to which we would defer. We think that there has been some shift in the style of scientific writing for the purposes of clarity and because English is a universal communication language, we want our writing to be easy to understand to a wide range of first and second language speakers. 

L97 – When was the accelerometer fitted? At recruitment? Was there a baseline activity collection period?

Authors' response: We fitted BOTH the accelerometer and Freestyle Libre at the same time; it was at the end of the one-month dietary acclimation period and at the beginning of the 96-hour data recording period. The 96-hour data recording period was at the PEAK OF SEASON ACTIVITY (see our previous comment for L88). So, there was no baseline recording period. We fitted the dogs with the devices, collected the blood (baseline samples), and off they went to work.

L126 – Serum tubes allowed to clot before what? Assuming this means allowed to clot before being put on ice since the samples were not processed on site, but please clarify.

Authors’ response: We added the word ‘Then’ (L136) and we hope that it is now clear: “Whole blood in the serum tubes was allowed to clot for 10 minutes whereas the whole blood in the sodium fluoride tubes was inverted several times to allow for the anticoagulant to mix with the blood. Then, the samples were stored on ice as long as the veterinarian was in the field and shipped on ice overnight to the principal investigator."

L136 – Remove comma after "whereas"

Authors' response: We removed the comma. 

L149-150 – Why no baseline measurement during month of acclimation?

Authors' response: There are two reasons for that. We simply did not have the resources and were working on a tight budget (we paid per the time the accelerometer recorded; also, each FSL sensor is quite expensive and has a maximum lifetime of two weeks, albeit it works for a much shorter time in most dogs). Also, it is one thing to ask the busy farmers to flash the readers over the sensors at least every 6 hours for 96 hours and a different thing to do so for a more extended period. Compliance, recruitment, and logistics were significant hurdles in this study. The second reason is that there was a big difference in the farm workload between the beginning of the acclimation period and the beginning of the 96-hour data recording period. We agree that it would have been interesting to have this data and make comparisons. But we do not think that not having it changed our ability to answer our study question.

L159-160 – Please include any relevant citations for established blood glucose levels

Authors' response: We inserted a citation from a veterinary clinical biochemistry book (L168) that has a population-based reference interval for canine blood glucose. 

L168 – Please provide inter- and intra-assay variation and sensitivity for this assay

Authors' response: We added the requested information (L178): “The assay’s lower limit of detection and range are 0.5 µU/L and 1-300 mU/L, respectively. The within run coefficient of variation (CV) at insulin concentrations of 27 mU/L and 65 mU/L are 1.05% and 3.9%, respectively. The between run CV at insulin concentrations of 27 mU/L and 65 mU/L are 4.07% and 6.83%, respectively.”

L171 – Suggest brief explanation of what HOMA-IR is. What is normal? What is expected in this population of dogs?

Authors' response: We added the following (L182): “The homeostatic model assessment of insulin resistance (HOMA-IR) is an epidemiological tool used in people to assess insulin resistance. High HOMA-IR levels indicate an increased resistance to insulin, and lower levels indicate an increased sensitivity to insulin’s action. The HOMA-IR was calculated as previously described [17, 18] according to the following formula: plasma insulin x serum glucose / 22.5.”

L174-177 – I do not see the purpose of this arbitrary distinction into activity levels, particularly with so few animals who were all highly active working dogs. This becomes a problem later on in the statistical analysis section when activity (a response variable) is used as a fixed effect for IG analysis.

Authors' response: stratification to three levels of activity allowed us to see the diet x activity level interaction on IG levels. As the Reviewer can see, there is a different metabolic ‘behavior’ at low/moderate levels of activity between the high and low CHO diets. This opens an intriguing question of why this is happening and what are the underlying physiologic mechanisms involved. Furthermore, as the Reviewer can see, the level of IG for the low/moderate activity levels in the high CHO diet is not different than that of the high level of activity in the low CHO diet. Reenforcing the argument above, it possibly suggests different utilization of energy resources for metabolism (i.e., fat vs. CHO) between the diets (and possibly between different levels of activity). Lastly, stratification of activity to three levels also allowed us to determine in what level of activity the dogs are more likely to have low glucose readings (an interesting question on its own).

L176 – Is the =< supposed to be ≤ or 

Authors' response: We changed the signs to be consistent with the Reviewers.

L198-199 – Were data transformations attempted?

Authors' response: Variables that did not follow normal distributions were analyzed with non-parametric tests to avoid the transformation and provide clear biological explanation on a nominal scale. We think that it is a valid statistical approach.

L201-203 – several questions:

- Why is time not included as a fixed effect?

Authors' response: Thank you for this important comment. Following your comment, we included ‘period’ as a fixed effect after converting the 96 hours into an 8-level categorical variable, ran the model again, and the only thing that changed is that there was no difference between the low and moderate levels of activity within diet 1 (high CHO low fat). Fig 3 was revised to reflect this change. This is now stated in L213-220.

- Above, you mention activity and inactivity periods, but these are also not reflected in this analysis

Authors' response: To prevent confusion, we changed ‘activity period’ and ‘inactivity period’ to ‘working period’ and ‘resting period’, respectively. The description for the statistical analysis of the frequencies of low vs. normal/high IG readings during the ‘working’ and ‘resting’ periods was explained in L241.

- What about farm, sex, age, breed, etc.? If these were evaluated as fixed effects then dropped, that should still be documented here.

Authors' response: We explored each one as a single effect and dropped from the final model because they were not significant.

- Activity is a response variable, not a fixed effect. If you wanted to include "activity level" as a fixed effect, you would need baseline activity data by which to block the dogs by activity level prior to diet allocation & acclimation

Authors' response: We did not analyze activity per se, but we used delta-g as a measure of activity. We think that it created a confusion (for which we apologize) and we corrected the manuscript to indicate that the dependent variables were IG and delta-g (not activity as you correctly indicated). The study was designed to assess the effect of diet on the level of IG and delta-g as a measure of activity. The classification by L/M/H activity levels enabled us to find biologic/metabolic changes in IG that are related to the interaction between diet x activity level.

In order to establish a ‘baseline’, we would have had to fit the dogs with accelerometers and FSL sensors at the beginning of the acclimation period, which as we explained above in response to one of your previous comments (L149-150) was not feasible financially and logistically. We think that the procedure to classify dogs as having low, moderate, and high activity levels was the best way we could classify the dogs. We think that our approach is valid for answering the research question.

L207-208 – Again, were any transformations attempted prior to analysis as a nonparametric variable?

Authors' response: Variables that did not follow normal distributions were analyzed with nonparametric tests to avoid the transformation and provide clear biological explanation on a nominal scale. We think that it is a valid statistical approach.

L208 – Remove comma after "whereas"

Authors' response: We removed the comma.

L209 – Why GLM here but MIXED above? What was included in the GLM model?

Authors' response: Thank you again for your comment. To be consistent, we changed the analysis from GLM to the MIXED procedure and described the MIXED model more precisely (L226).

L213 – Why not include time & diet x time interaction here?

Authors' response: Thank you for the suggestion. We modeled the curves of activity and IG through the time for each dog. This description of the model was modified in the text. To further explain, the modeling of the curves for level of glucose and the delta-g were done at the level of the dog. Later, we wanted to obtain the average curve of level of glucose and delta-g for each of the diets, which was achieved by averaging individual dog’s splines coefficients to achieve the mean and standard error of the coefficients for each diet. The interaction between diet and time can be evaluated but it is difficult as in this case time is a continuous variable. From Fig 2 it is seen that the level of glucose is different across time for each of the diets. If diet and time interaction would have been tested, they would have been significant. However, it was hard to achieve it because the use of splines. If the modeling of the dependent variables IG and delta-g would have been done with a polynomial the interaction could have been achieved but we chose not to use a polynomial because a polynomial will not describe adequately the variation for these variables.

L214 – Is "activity level" here referring to the "activity levels" described above (L174-177)? Why examine activity level as a dependent variable here but a fixed effect in the linear models above?

Authors' response: We apologize for the confusion. We changed the text to IG and delta-g which refers to the dependent variables that were modeled.

Results:

L221 – How was BW statistically analyzed? This is not described in methods. The description of BW effects is a bit clunky. Were diet and time fixed effects significant? The interaction? You jump straight to describing the individual diet x time interactions without presenting results of main effects.

Authors' response: We added the description of analysis of BW in the methods L232 and the results are presented in the result section L259.

L225 – Suggest authors use consistency with respect to P-value reporting – 2 decimals used here versus three in next line. Recommend confirming if journal has standard for reporting P-values.

Authors' response: We changed all P values to include 3 decimal places.

L226 – With the standard errors reported here, these weights should not have a statistical difference. Something is not correct with this analysis.

Authors' response: Thanks for this observation which led us to review and revise this part of the statistical analysis in the statistical methods. The LSM were similar but the standard error changed which make the results significantly different.

L229 – What about time effect? Diet x time interaction?

Authors' response: Thanks for this important point! The modeling of the curves for level of glucose and the delta-g were done at the level of the dog. Later, we wanted to obtain the average curve of level of glucose and delta-g for each of the diets, which was achieved by averaging individual dog’s splines coefficients to achieve the mean and standard error of the coefficients for each diet. The interaction between diet and time can be evaluated but it is difficult as in this case time is a continuous variable. From Fig 2 it is seen that the level of glucose is different across time for each of the diets. If diet and time interaction would have been tested, they would have been significant. However, it was hard to achieve it because the use of splines. If the modeling of the dependent variables IG and delta-g would have been done with a polynomial the interaction could have been achieved but we chose not to use a polynomial because a polynomial will not describe adequately the variation for these variables. 

L242 – Your hypothesis is that low CHO diets lead to low IG which in turn reduces activity. Now you are trying to attribute activity to IG levels! Again, activity is a response variable, not a fixed effect. Dogs' activity is likely related to IG, but that is not what this study was designed to assess. This section should be removed. “activity is a response variable, not a fixed effect” 

Authors' response: We did not analyze activity per se, but we used delta-g as a measure of activity. We think that it created a confusion and we corrected the manuscript to indicate that the dependent variables were IG and delta-g (not activity as you correctly indicated). The study was designed to assess the effect of diet on the level of IG and delta-g as a measure of activity. The classification by L/M/H levels of activity enabled us to find biologic/metabolic changes in IG that are related to the interaction between diet x activity level.

L256 – Period of the day is not included in your statistical analysis (L201-203) as a fixed effect.

Authors' response: Good observation. We revised the description of the statistical analysis of the frequencies of low vs. normal/high IG readings during the ‘working’ and ‘resting’ periods, which is now explained in L241. We looked at frequencies of low vs. norm/high IG and used the Chi Square test to determine if there were differences.

L256 – Is the incidence of low IG more relevant than time spent at low IG? How much total time was spent at low IG? The potential diet x time interaction is important.

Authors’ response: The incidence of low glucose is important as well as the overall time of low IG. The incidence is important because until recently when veterinarians started to use the FSL (FreeStyle Libre) we did not think that healthy animals have low IG readings. We are saying it carefully because we do not have reference intervals for IG (as we discussed it in length in the discussion). The point is that now we know (from this study and growing clinical experience) that healthy animals could have low IG readings without clinical signs of neuroglycopenia. How many times a dog has low IG is important because its sympathoadrenal responses should prevent it from happening too often. If the incidence starts to increase above a certain level (which is yet to be defined) then there might be a physiologic or clinical problem. However, the ‘normal’ incidence and frequency of low IG in dogs is unknown and this study would be the first to show it. At the same note, how long a dog has low levels of IG is also important. If there are prolonged periods of low IG, the probability of development of clinical signs related to neuroglycopenia increases as well as the chance that an underlying physiologic or clinical problem exist. 

In L312 we described that median (range) of low IG events is 6.5 (IQR 17.5; range 1-42). A median of 6.5 equals to 6.5 x 15 min = 97.5 min per the 96-hour of data recording. You can see that one dog spent 10.5 hours (not continuously) per the 96 hours of data recording with low IG level. This dog as much as we know has been completely normal and did not have any clinical problems or changes in its quality of work (in NZ working dogs are “decommissioned” if they have a decrease in work quality).

We analyzed the interaction of time x diet when IG is low by doing a Chi-Square test and found it to be significant (p < 0.001). The result is now indicated in the text in L308. 

L266 – Remove extra space between "Diet 2" and "group"

Authors’ response: Extra space removed.

L269-270 – Would it be expected for the majority of low IG readings to occur in the activity period? Would be nice to mention whether or not this was expected.

Authors’ response: Yes, we expected that most low IG would be during the working period. We prefer to discuss our expectations and interpretation of the results in the discussion rather than in the result section.

L275 – Remove comma after "whereas"

Authors’ response: Extra space removed.

L281 – Again, activity is not a fixed effect, and your hypothesis has the reverse – low IG reducing activity. This section should be removed.

Authors' response: We did not analyze activity per se, but we used delta-g as a measure of activity. We think that it created a confusion and we corrected the manuscript to indicate that the dependent variables were IG and delta-g (not activity as you correctly indicated). The study was designed to assess the effect of diet on the level of IG and delta-g as a measure of activity. The classification by L/M/H activity levels enabled us to find biologic/metabolic changes in IG that are related to the interaction between diet x activity level.

L299 – Consider re-wording as there is no such thing as "inappropriate" serum insulin.

Authors’ response: The sentence was changed to “None of the dogs had evidence of hyperinsulinemic hypoglycemia…” (L338)

Discussion:

L304 – These results were presented above, the reference to the figure is not needed here.

Authors’ response: We started the discussion with the statement of the most important finding in the study and developed the paragraph over this statement. It was specifically requested by Reviewer 1 during the previous revision and we made extensive changes in the structure of the discussion to accommodate Reviewer 1’s request. We think that it is legitimate to refer to the figures in the discussion. If the editor would require removing the reference to the figures in the discussion, we will remove it. 

L307 – Suggest removing "most"

Authors’ response: We removed ‘most’

L314 – Reword "people" as "humans" throughout

Authors’ response: The word ‘humans’ substituted the word ‘people’ throughout the manuscript.

L321-322 – Where is it suggested that low fat/high CHO diet offers advantages? Your hypothesis is that lower CHO/high fat diet will reduce activity.

Authors’ response: The word ‘above’ in L360 refers to the beginning of this paragraph where we discuss the advantages of adaptation to a high fat low CHO diet.

L323-326 – These two sentences are very repetitive. Suggest condensing into one sentence.

Authors’ response: We condensed the two sentences to one (L362) “For example, exhaustive exercise in sled dogs that consumed a diet with 0% of metabolizable energy from a CHO source and high %ME from fat had higher albumin, total calcium, and magnesium than control dogs on diets with higher %ME from a CHO source [27].”

L334 – Reword to "necessarily"

Authors’ response: The word ‘necessary’ was removed.

L339-340 – Here you could discuss how activity could have an impact on IG and recommend additional research since your study was not designed to examine this effect.

Authors’ response: At the end of the paragraph we added the following L385: “However, further studies are required to provide a mechanistic insight to determine whether either or both of the above explanations, or alternative explanations underly the temporal relationship between delta-g and IG as our study was not designed to investigate it.”

L340 – Remove reference to figures from discussion

Authors’ response: We think that it is legitimate to refer to the figures in the discussion. We will refer to the Editor’s decision about this point.

L347 – Suggest further discussion at the end of this paragraph as to the potential significance of this observed relationship.

Authors’ response: We do not think that in isolation, the temporal relationship that we described in this paragraph, has a significant implication as both 1) increased glucose production by the liver and reduced glucose utilization by insulin-dependent tissues at high activity levels, and 2) increased muscle and liver glucose uptake immediately after low activity levels have been described in dogs. We think that the pronounced fluctuation in IG and a lower AUC glucose in Diet 2 vs. Diet 1, combined with the fact that dogs in Diet 2 had higher AUC delta-g is really the interesting finding here that has a biological significance and we discussed it in the first paragraph.

L349-350 – Awkward wording, suggest re-working. Since the hypothesis is that low IG leads to reduced activity, suggest focusing on that. May discuss potential for higher activity to reduce IG as avenue for future research, but not really relevant here.

Authors’ response: The sentence was rephrased as follow (L390): “During the 96 h of data recording there were neither low peaks of IG that followed high peaks of delta-g, nor low peaks of delta-g followed low peaks of IG.” What we are saying is that the temporal relationship seen in Fig 2 was that low IG always followed low delta-g and that we think that it was because of the large flux of glucose into the liver and muscle as dogs started to rest after high activity level. We discussed it in the previous paragraph and this paragraph basically indicate that they did it in high efficiency because they were very sensitive to insulin. 

L351 – Expand on the potential limitation of the 96 h collection period. What time period would have been better?

Authors’ response: This is a hard question to answer and we do not have an answer. It maybe never if the explanation that we gave is correct (i.e., low IG followed low delta-g because large flux of glucose into the liver and muscle occurred as dogs started to rest immediately after periods of high activity level) or a slightly longer or much longer than 96 h if low peaks of IG would follow high peaks of delta-g, or low peaks of delta-g would followed low peaks of IG. 

L358 – Suggest "of this alternative"

Authors’ response: We made the requested suggestion (L395)

L360-361 – Suggest removing "The incidence of low IG episodes was 3.12% per 96 h and involved 10/22 (45.5%) of the dogs" from the discussion since it has already been reported in results section.

Authors’ response: We deleted the sentence.

L363 – Similar to comment in L299 above, please remove "inappropriate" and re-word.

Authors’ response: L401 now reads: “The dogs in this study had their serum insulin and glucose concentrations measured at the beginning and the end of the 96 h study periods, and none of the dogs had results that were suggestive of hyperinsulinemic hypoglycemia.”

L364 – Suggest removing "we conclude that"

Authors’ response: The sentence now reads as follows (L404): “Therefore, healthy dogs could have a low incidence of low IG and the frequency of low IG may be greater in dogs fed ultra-low CHO high fat diet (Diet 2).”

L365 – Is the fact that healthy, active dogs can have low IG really a surprising conclusion/finding of this study? Suggest writing different sentence to link discussion between serum insulin/glucose and IG

Authors’ response: Yes, it was a surprise. Having low levels of glucose of this magnitude has been thought of as an abnormal or pathological condition. Seeing that normal healthy dogs could have it on occasion changes the perspective about finding low IG in dogs. As we eluded, low IG ≠ low blood glucose (BG) especially when you do not have reference intervals. Apparently, there could be many things that can induce low IG and this study identified diet and activity as two factors that were related to it. 

L373 – What effect might feeding time/frequency have on IG?

Authors’ response: Interesting question! If dogs are not adapted to alternative sources of energy (i.e., fat) then we speculate that low glucose levels would be more likely to happen as the time elapsing between the incidence of low glucose and feeding increases. This could aggravate if the dog is under any chronic stress (physical, pathological, mental). On the other hand, low levels of glucose could also happen postprandially if the incretin-mediated insulinemic effect is not met with a matching opposing effect mediated by glucagon ± cortisol.

L377 – How does the evening feeding time play into this? Suggest further discussion

Authors’ response: We do not know the answer to this question. We explain just above how low levels of glucose might happen postprandially. But there was a strong diet related effect, so it is not simply a timing issue. We think that we discussed it adequately in this paragraph.

L409 – Any suggestions for future investigation regarding IG monitoring? Potential future work could be worked in throughout discussion.

Authors’ response: To know, one would need to perform invasive studies in dogs that are way beyond the scope of this study. These would include catheterization of the portal and hepatic veins to measure pattern of pancreatic hormone secretion and hepatic glucose output in dogs that will be on high CHO low Fat vs. low CHO high fat combined with EEG monitoring of sleep patterns, IG monitoring, BG monitoring, and VO2 monitoring in combination with periods of strenuous activity (on a treadmill or alike) and rest. This is just not something you can achieve with a simple field study such as this.

L418 – Suggest addition of how this limitation could be overcome, such as formulating the diets to be similar in every respect except for source of energy (CHO vs. fat)

Authors’ response: We added the following in L457: “A future study could reduce nutritional confounders by formulating the diets using varying proportions of the same ingredients, such that only the macronutrient proportions differ.”

Conclusion:

L435 – Spell out abbreviation when it begins a sentence.

Authors’ response: (L476) we spelled out interstitial glucose. 

Tables and Figures:

Fig. 1 – in caption, should read "an ultra-low CHO…" not "a"

Authors’ response: (L85) we revised to "an ultra-low CHO…"

Table 2 – Diet randomization is clearly not balanced across farms, particularly farm 3. If there is a reason for this, please include in methods section.

Authors’ response: We did our best to balance the diets across the four farms. In farm 3 there was a problem and it was not as balanced as in the other farms. Obviously now we cannot do anything about it but looking at the variables in Tables 1 and 2 one can tell that the randomization process was successful because there were no significant differences in the demographics across farms and diets.

Table 3 – Recommend removal since activity should not be used as a fixed effect.

Authors’ response: We assume the Reviewer meant Fig 3 as the manuscript does not have Table 3. We explained above that we did not analyze activity per se, but we used delta-g as a measure of activity. We think that it created a confusion and we corrected the manuscript to indicate that the dependent variables were IG and delta-g (not activity as you correctly indicated). The study was designed to assess the effect of diet on the level of IG and delta-g as a measure of activity. The classification by L/M/H levels of activity enabled us to find biologic/metabolic changes in IG that are related to the interaction between diet x activity level. Fig 3 is important because it shows the interaction between activity level and diet with respect to IG. As we explained before, the stratification to three activity levels allowed us to see the diet x activity level interaction on IG levels. There was a different metabolic ‘behavior’ at low/moderate levels of activity between the high and low CHO diets. This opens an intriguing question of why this is happening and what are the underlying physiologic mechanisms involved. Furthermore, as the Reviewer can see, the level of IG for the low/moderate activity levels in the high CHO diet is not different than that of the high level of activity in the low CHO diet. Reenforcing the argument above, it possibly suggests different utilization of energy resources for metabolism (i.e., fat vs. CHO) between the diets (and possibly between different levels of activity). Lastly, stratification of activity to three levels also allowed us to determine at what activity level the dogs are more likely to have low IG readings (an interesting question on its own). We request to keep the figure from these reasons.

---

## [Editor Report · Decision Letter 2]

6 Dec 2021

Less is more? Ultra-low carbohydrate diet and working dogs’ performance

PONE-D-21-02404R2

Dear Dr. Arnon Gal,

We’re pleased to inform you that your manuscript has been judged scientifically suitable for publication and will be formally accepted for publication once it meets all outstanding technical requirements.

Kind regards,

Balamuralikrishnan Balasubramanian

Academic Editor

PLOS ONE
---

## [Editor Report · Acceptance letter]

14 Dec 2021

PONE-D-21-02404R2 

Less is more? Ultra-low carbohydrate diet and working dogs’ performance 

Dear Dr. Gal:

I'm pleased to inform you that your manuscript has been deemed suitable for publication in PLOS ONE. Congratulations! Your manuscript is now with our production department. 

Kind regards, 

on behalf of

Dr. Balamuralikrishnan Balasubramanian 

Academic Editor

PLOS ONE